# Quality of life and cognitive assessment in healthy older Asian people with early and moderate chronic kidney disease: The NAHSIT 2013–2016 and validation study

**Sheng-Feng Lin**[1,2,3], **Yen-Chun Fan**[1], **Tzu-Tung Kuo**[1], **Wen-Harn Pan**[4], **Chyi-Huey Bai** [1,2,5]*

**1** Department of Public Health, School of Medicine, College of Medicine, Taipei Medical University, Taipei, Taiwan, **2** Department of Public Health, College of Public Health, Taipei Medical University, Taipei, Taiwan, **3** Department of Emergency Medicine, Taipei Medical University Hospital, Taipei, Taiwan, **4** Institute of Biomedical Sciences, Academia Sinica, Taipei, Taiwan, **5** Nutrition Research Center, Taipei Medical University Hospital, Taipei, Taiwan

* baich@tmu.edu.tw

**Data Availability Statement:** The data is not publicly available. The NAHSIT 2013–2016 study was managed by the Health Promotion and

## Abstract

### Background

Taiwan has the highest prevalence of chronic kidney disease (CKD). Impaired cognition and quality of life are significant phenomena in the late stages of CKD. We sought to obtain an overview and the attributable effect of impaired glomerular filtration on multiple domains in cognition and dimensions of quality of life for community-based healthy older adults in Taiwan.

### Methods

The study was derived from the Nutrition and Health Survey in Taiwan (NAHSIT) 2013–2016, a nationwide cross-sectional study conducted to sample healthy, community-based older adults aged $\geq$65 years in Taiwan. Participants were categorized into four CKD groups: CKD stage 1, stage 2, stages 3a and 3b, and stages 4–5. The Mini-Mental State Examination (MMSE) and the QoL questionnaire derived from the 12-item Short Form Health Survey (SF-12) were measured. Generalized linear mixed models (GLMMs) and principal component regressions were employed for the analysis and validation, respectively.

### Results

Participants with moderate CKD (stages 3a and 3b) showed deficits in global MMSE, domain orientation to time, calculation, complex commands, and role-physical and vitality in QoL questionnaires. In GLMMs, impaired eGFR per 30 mL/min/1.73 m$^2$ was associated with lower global MMSE scores (β = -0.807, standard error [SE] = 0.235, P = 0.0007), domain orientation to time (β = -0.155, SE = 0.047, P = 0.0011), calculation (β = -0.338, SE = 0.109, P = 0.0020), complex commands (β = -0.156, SE = 0.079, P = 0.0494), and role-physical (β = -2.219, SE = 0.779, P = 0.0046) dimensions of QoL.

Administration (HPA), Ministry of Health and Welfare, Taiwan. The website of the Health Promotion and Administration (HPA), Ministry of Health and Welfare, Taiwan offers an electronic system of contact [https://www.hpa.gov.tw/EngPages/EngOpinionMailbox.aspx]. With legal restrictions imposed by the government of Taiwan on the distribution of the personal health data in relation to the "Personal Information Protection Act," requests for data need a formal proposal which should be directed to the Health and Welfare Data Science Center (HWDC), Ministry of Health and Welfare, Taiwan.

**Funding:** CH Bai received the funding. This work was funded by the Health Promotion Administration, Ministry of Health and Welfare, Taiwan (MOHW109-HPA-H-114-144702) and Ministry of Science and Technology, Taiwan (MOST 110-2314-B038-056-MY3). The content of this research may not represent the opinion of the Health Promotion Administration, Ministry of Health and Welfare. The funder had no role in the study design, data collection and analysis, decision to publish, or preparation of the manuscript.

**Competing interests:** The authors have declared that no competing interests exist.

## Conclusions

Elderly Han Chinese adults with moderately impaired renal filtration could manifest cognitive deficits in orientation to time, calculation, and impaired quality of life in physical role functioning.

## Introduction

### High prevalence of chronic kidney disease in Taiwan

In Taiwan, the incidence rate and prevalence throughout all stages of kidney disease were approximately 27,210 per million person-years and 154,600 per million people in the last decade [1]. According to the United States Renal Data System (USRDS) annual report in 2020 [2], Taiwan has the highest prevalence of end-stage renal disease (ESRD) (3,587 per million population) and the highest prevalence of dialysis (3,429 per million population) worldwide [3, 4]. In addition, the USRDS in 2020 [2] reported that the prevalence of ESRD is highest in the age group of 45–64 years (25,156 per million population), compared to 20–44 years (5,290 per million population), 65–74 years (17,194 per million population), ≥75 years (15,066 per million population). Due to improved management for CKD patients, life expectancy [5, 6] and social burden of care [7] are both expected to increase in Taiwan.

### Measures of chronic kidney disease

Renal disease is assessed by the degree of injured kidney filtration rate [8, 9], and CKD is relatively asymptomatic until the end stage [10]. The most commonly used measure is estimated glomerular filtration rate (eGFR) by serum creatinine, which evaluates the degree to which the kidneys remove waste from the blood. According to eGFR, CKD is divided into six categories by the international guideline group Kidney Disease Improving Global Outcomes (KDIGO) [11]: early CKD of stage 1 (eGFR of > 90 mL/min/1.73 m$^2$), stage 2 (eGFR of 60–89 mL/min/1.73 m$^2$), stages 3a (eGFR of 45–59 mL/min/1.73 m$^2$), and 3b (eGFR of 30–44 mL/min/1.73 m$^2$), late CKD stage 4 (eGFR of 15–29 mL/min/1.73 m$^2$), and stage 5 ESRD (eGFR < 15 mL/min/1.73 m$^2$). In addition, eGFR is modified by age, sex, and ethnic group [12–14].

### Cognitive impairment in chronic kidney disease

Theoretically, the brain and kidney have similar microvascular structures and are susceptible to hemodynamic fluctuations [15–17]. Recent studies suggested that cognitive impairment is associated with cardiovascular risk factors, such as hypertension, hyperlipidemia [18, 19], and diabetes mellitus [20, 21]. Similar to the brain, the kidney receives a large amount of blood flow: 1.0–1.2 liters per minute per 1.73 m$^2$ of body surface area, approximately 20–25% of the total cardiac output [22]. When renal function deteriorates, uremic toxins and waste from blood are not removed, and the build-up of toxins impairs brain function [23]. A recent study [24] showed that CKD mainly affects the cortical synchronization of neurons in electroencephalography. Dementia is recognized as a significantly common phenomenon in ESRD, affecting as high as 40% of patients [25]. Studies have reported that ESRD patients treated with hemodialysis exhibited cognitive impairment in the domains of orientation, attention, memory, construction, and executive function [26, 27]. Additionally, ESRD patients who received kidney transplant showed a higher incidence of dementia [28]. While the vast majority of studies which investigated patients with both renal function injury and cognitive function focused

on late CKD and ESRD populations, to date, few studies have evaluated cognitive impairment in early CKD stages [29].

Mild cognitive impairment (MCI), an intermediate cognitive state between normal and full-blown dementia, is characterized by problems with orientation, attention, judgment, and memory [30, 31], and has been reported to be associated with cardiovascular risk factors in the last two decades. Recently, there have been increasing studies supporting the association between CKD and MCI. Two recent meta-analyses [32, 33] showed that impaired eGFR is associated with cognitive impairment and can be present in the early and late stages of CKD. A systematic review [29] included three relevant studies on CKD patients <65-years-old, which found that these patients had slow processing speed [34], impaired verbal learning [35, 36], and impaired working memory [35] in moderate stages of CKD. A survey of Japanese CKD patients [37] demonstrated that female participants with eGFR <90 mL/min/1.73 m$^2$ showed a faster rate of cognitive decline than those with eGFR $\geq$90 mL/min/1.73 m$^2$.

## Quality of life in chronic kidney disease

The term "quality of life" can be traced back to the definition in 1948 by the World Health Organization (WHO) as a state of perception of "physical, mental, and social well-being, and not merely the absence of disease" [38]. Prior meta-analysis [39] assessed the utility-based quality of life with the 12-item or 36-item Short Form Health Survey (SF-12 or SF-36), Euro-Qol Group's EQ-5D, and other instruments, which suggested that dialysis is significantly associated with reduced quality of life for ESRD patients. However, few studies have investigated the association between quality of life and the early and moderate stages of CKD. We found that these studies exhibited inconsistent results: (1) a cross-sectional study of $\geq$60 year-old participants in North California [40] showed no relationship between eGFR and quality of life scores in physical and mental dimensions after adjustment for medical comorbidities; (2) a cross-sectional study including $\geq$50 year-old Irish participants [41] showed that reduced scores in quality of life are associated with eGFR by serum cystatin rather than creatinine; and (3) a recent cross-sectional study [42] enrolled 2,255 older adults in European countries and found that early CKD at stages 3a and 3b, defined by the BIS equation, is associated with lower quality of life on the EQ-Visual Analogue Scale (EQ-VAS). To our knowledge, there is a lack of relevant research on ethnic Han Chinese as well.

## Study rationale and aims

Cognitive impairment or dementia is associated with increased physical disability [43, 44] and worsened caregiver burden [45]. While The World Health Organization (WHO) defined a society in which the proportion of people aged $\geq$65 years is 7% or higher is known as an "aging society," 14% or higher is regarded as an "aged society," and the prevalence of older adults in Taiwan exceeded 14% in 2018. Since Taiwan has become an aged society and has the highest worldwide prevalence of CKD and ESRD [2], assessment of cognition in CKD is of paramount importance. In this study, we hypothesized that the high prevalence of cognitive impairment in older adults with CKD in Taiwan, and perception of the physical and mental well-being of older adults was worsened in the early and moderate stages of CKD. In this cross-sectional study, we aimed to (1) obtain an overview of cognitive impairment and quality of life at each stage of CKD in the Han Chinese population, and (2) assess the attributable effect of poor kidney filtration function, after adjusting for demographic factors, on multiple cognitive domains and dimensions of quality of life among community-based healthy older adults in Taiwan. Additionally, we performed a validation study to examine the robustness of the association between eGFR, cognitive impairment,

and quality of life. The existence of collinearity in our data analysis was likely since multiple domains of cognition and multiple dimensions of life quality were assessed. We replaced the conventional regression models with partial least squares (PLS) regressions to deal with potential collinearity and confirm robustness.

## Methods

### Study design and participants

The Nutrition and Health Survey in Taiwan (NAHSIT) 2013–2016 [46] was a nationwide cross-sectional study in Taiwan, which was conducted from January 1, 2013 to December 31, 2016. The primary objective of conducting NAHSIT 2013–2016 by the Health Promotion and Administration, Ministry of Health and Welfare in Taiwan was to determine the nutritional status of the Taiwanese population, which was used as a reference for nutrition and health policymaking in a government entity. The NAHSIT involved a systematic and stratified sampling process that covered all 359 townships and city districts in Taiwan, and was divided into eight strata according to population density and geographical area. The content of the survey included (1) behavioral indicators, such as cognition and quality of life assessment; (2) health outcome indicators, such as cardiovascular disease and kidney disease; and (3) laboratory tests of clinical chemistry, such as serum creatinine, lipid profiles, and blood glucose. Data collection was for the primary objective of the study. All participants were visited once during the study period. Our study was a secondary analysis of data derived from NAHSIT 2013–2016. Data for participants aged ≥65 years who completed the MMSE and QoL questionnaire of the 12-item Short Form Health Survey (SF-12) were collected. Participants who had no laboratory examination of renal function were excluded. This study was approved by the Institutional Review Board of Biomedical Science Research, Academia Sinica, Taiwan (AS-IRB01-13067) and the Research Ethics Committee, National Health Research Institutes, Taiwan (EC1020110). Written informed consent was obtained from all participants.

### Cognitive and life quality measures

In Taiwan, the most widely used cognitive assessment tool is the culturally adapted traditional Chinese version of the Mini-Mental State examination (MMSE) [47]. The norms have been validated in the Taiwanese population [48] and other Han Chinese populations [49, 50]. The traditional Chinese version of the MMSE includes questions on orientation to time (5 points), orientation to place (5 points), registration (3 points), calculation (5 points), memory recall (3 points), language (2 points), repetition (1 point), and complex commands (6 points), and the MMSE score ranges from 0 (worst) to 30 (best) [51, 52]. For the Taiwanese population, the norms [48] indicated 2 cutoff scores for determining cognitive impairment: a value of <27 for the literate and another value of <16 for the illiterate, respectively.

The most well-validated tool for assessment of quality of life for the Taiwanese population is the Traditional Chinese version of the SF-36 [53, 54]. Despite being valid and equivalent for Han Chinese and other ethnic groups, SF-36 is limited by its length and time-consuming characteristics. The traditional Chinese version of SF-12, an abbreviated form of SF-36, is validated as equivalent to SF-36 for assessing the dimensions of physical and mental health [55]. The SF-12 questionnaire includes five dimensions of concepts: role-physical (RP), vitality (VT), social functioning (SF), role-emotional (RE), and mental health (MH). Each concept was transformed into a scale ranging from 0 to 100, with a mean distribution of 50 and a standard deviation of 10. A higher score signifies a perception of a better quality of life [56, 57].

## Data collection process

Door-to-door visits were conducted by trained interviewers to collect demographic characteristics; culturally adapted traditional Chinese version of the MMSE [48] and SF-12 were administered. After informed consent was obtained from participants, phlebotomy was conducted for laboratory assessments of blood urea, nitrogen, creatinine, blood glucose, HbA1c, and lipid profiles of total cholesterol, low-density lipoprotein cholesterol (LDL-C), high-density lipoprotein cholesterol (HDL-C), and triglycerides. Blood pressure measurements were performed by trained public health nurses and after participants rested for 5 minutes with the arm placed at the same height as the heart. A standard sphygmomanometer with an appropriate cuff size was used. Blood pressure measurements were recorded three times, and a fourth measurement was performed if two of the three BP measurements varied by $\geq$ 10 mmHg. Diabetes mellitus was defined as fasting glucose $\geq$126 mg/dl or HbA1c level $\geq$6.5%, in accordance with the standard of the American Diabetic Association [58]. Hypertension was defined as systolic blood pressure (SBP) $\geq$140 mmHg or diastolic BP (DBP) $\geq$90 mmHg according to the American Heart Association standard [59].

## Definition of chronic kidney disease groups

A previous study [60] indicated poor agreement between the modification of diet in renal disease (MDRD) and the Cockcroft-Gault equations for assessing CKD in cognitively impaired older adults. In Taiwan, citizens are mono-ethnic, and approximately 97% of the Taiwanese are Han Chinese [61]. For older Chinese adults, two studies [62, 63] concluded that eGFR derived from the Berlin Initiative Study (BIS) equation achieved superior accuracy and lower rates of misclassification of CKD stages compared to the MDRD and Chronic Kidney Disease Epidemiology Collaboration (CKD-EPI) equations. Because the enrolled participants were healthy older adults in the NAHSIT 2013–2016, eGFR was obtained by the BIS equation (eGFR = 3736 × creatinine$^{0.87}$ × Age$^{0.95}$ × 0.82 [if female]) [64, 65]. All participants were categorized into three CKD groups according to the derived eGFR: CKD stage 1 (group 1; eGFR > 90 mL/min/1.73 m$^2$), stage 2 (group 2; eGFR of 60–89 mL/min/1.73 m$^2$), stages 3a and 3b (group 3; eGFR of 30–59 mL/min/1.73 m$^2$), and stages 4–5 (group 4; eGFR of 0–29 mL/min/1.73 m$^2$). Since our study targeted the healthy older population, the number of late CKD stages 4–5 was expected to be small, and therefore Stages 4 and 5 were grouped into one category. In addition to eGFR, we confirmed the CKD risk group stages by urine albumin-to-creatinine ratio (ACR). Compromised renal function allows albumin to pass into the urine. According to the KDIGO [11], urine ACR <30 mg/g is defined as normal to mildly increased albuminuria, urine ACR of 30–300 mg/g as moderately increased albuminuria, and urine ACR >300 as severely increased albuminuria.

## Statistical analysis

Demographic characteristics, eGFR, scores on each domain of MMSE, and SF-12 among the four CKD groups were assessed with one-way analysis of variance for continuous variables and with chi-squared or Fisher's exact test for categorical variables. The *P* trend for the four CKD groups was estimated using generalized linear models for continuous variables and the Cochran-Armitage trend test for categorical variables. Generalized linear mixed models (GLMMs), containing both fixed and random effects, were applied for data analyses. Because of the study design, which involved a systemic and stratified sampling from the nationwide population in Taiwan and measurements of multiple related factors, a subject-specific model with a random intercept was used. We described this model in the following matrix formula $Y_i = X_i\beta + R_i + E_i$, where $Y_i$ denotes the matrix containing a set of measures of all items of the

MMSE and SF-12 on the $i$th older adult participant as dependent variables, $X_i$ denotes the matrix containing a set of intercepts, independent variables of eGFR (per 30 mL/min/1.73 m$^2$), and the covariates, $\beta$ denotes the vector of estimated regression parameters, $R_i$ denotes an intercept random effect associated with *the i*th older adult participant, which has a normal distribution and is independent and identically distributed (i.i.d.); and $E_i$ denotes random error. The covariates used in the GLMMs were none (model 1), a set of variables with age, sex, and interaction of age and sex (model 2), a set of variables with age, sex, and interaction of age and sex, and education years (model 3), and a set of variables with age, sex, and interaction of age and sex, education years, and the characteristics that were imbalanced (defined as $P < 0.05$) among the four CKD groups (Model 4). All statistical analyses were performed using SAS software (version 9.4; SAS Institute, Cary, NC, USA). Applying the Bonferroni correction with the formula $p \leq \alpha/m$, where $\alpha$ was defined as 0.05, and the number of groups for multiple comparisons was 4. In our analysis, a two-tailed $p$ value $< 0.0125$ (0.05/4) was considered statistically significant.

## Validation study

To verify the association between renal function injury, cognitive impairment, and quality of life, we swapped the factors of eGFR in older adults and measured all items of the MMSE and SF-12 in the regression analysis. Conventionally, all items of the MMSE and SF-12 were employed as dependent variables (or response variables), and eGFR was used as an independent variable (or explanatory variable). Instead, we conducted partial least squares (PLS) regressions with the following formula: $Y = XB + B_0$, where $Y$ denotes the response matrix of eGFR, $X$ denotes the factor matrix of all items of the MMSE and SF-12, $B$ denotes the matrix of factor loadings, *and $B_0$* denotes error terms. Spearman's rank correlation tests were used to examine the correlations among all items of the MMSE and SF-12. One major advantage of PLS regression is that it can easily manage the multicollinearity problem in conditions when there are numerous highly correlated explanatory variables [66, 67]. PLS regressions can help identify the optimal linear combinations of explanatory variables by projecting them to new spaces. The PLS regression was performed using the PROC PLS procedure implemented in SAS 9.4 software, with a factor number of 2 and principal components regression (PCR) to obtain factor loadings. Variables with a factor loading of $> |0.3|$ were considered significant [68].

## Results

### Participant characteristics

A total of 497 participants were enrolled in this study. Their mean age was 72.2 ± 6.2 years, 213 (42.9%) were female, and the mean score of global MMSE was 26.8 ± 3.4. The prevalence of MCI was 12.7%. The demographic characteristics and laboratory test results of the four CKD groups are shown in Table 1. Of the included participants, 66 (13.3%) had CKD stage 1, 303 (61.0%) had CKD stage 2, 119 (23.9%) had CKD stages 3a and 3b, and 9 (1.8%) had CKD stages 4–5. In groups 1, 2, 3 and 4, the average age evaluations were 68.5 ± 3.8, 71.1 ± 5.2, 76.9 ± 6.6, and 74.3 ± 8.2 years, respectively ($P < 0.0001$); proportions of female participants were 69.7%, 42.9%, 27.6%, and 44.4%, respectively ($P < 0.0001$); and eGFR values were 100.3 ± 7.9, 74.2 ± 8.6, 47.7 ± 8.1, and 23.9 ± 4.6 mL/min/1.73 m$^2$, respectively. Regarding sociodemographic characteristics, years of education were lower in groups 3 and 4 ($P = 0.0023$), and individual income was not significantly different among all CKD groups ($P = 0.1435$). Medical comorbidities, including hypertension, hyperlipidemia, and diabetes, were not significantly different among the four groups. The systolic and diastolic blood

**Table 1. Renal function and mental function health status (N = 497).**

| CKD Staging | CKD stage 1 (N = 66) | CKD stage 2 (N = 303) | CKD stage 3a, b (N = 119) | CKD stage 4–5 (N = 9) | P value | P trend |
|---|---|---|---|---|---|---|
| Demographic characteristics | | | | | | |
| Age (years) | 68.5 ± 3.8 | 71.1 ± 5.2 | 76.9 ± 6.6 | 74.3 ± 8.2 | <0.0001* | <0.0001* |
| Female Sex (%) | 46/ 66 (69.7%) | 130/ 303 (42.9%) | 33/ 123 (27.6%) | 4/ 9 (44.4%) | <0.0001* | <0.0001* |
| Sociodemographic status | | | | | | |
| Education years | 10.2 ± 4.6 | 9.1 ± 4.9 | 7.6 ± 4.8 | 8.4 ± 3.1 | 0.0023* | 0.0003* |
| Annual income US$, n (%) | 6315.8 ± 4643.4 | 10484.7 ± 10666.9 | 8388.2 ± 8973.6 | 5800.0 ± 4764.5 | 0.1435 | 0.5001 |
| Kidney function measures | | | | | | |
| Blood urea nitrogen | 13.8 ± 3.3 | 15.6 ± 3.7 | 22.2 ± 6.1 | 39.0 ± 10.3 | <0.0001* | <0.0001* |
| Creatinine | 0.6 ± 0.1 | 0.8 ± 0.2 | 1.3 ± 0.3 | 2.9 ± 1.0 | <0.0001* | <0.0001* |
| eGFR (mL/min/1.73 m$^2$) | 100.3 ± 7.9 | 74.2 ± 8.6 | 47.8 ± 8.1 | 23.9 ± 4.6 | <0.0001* | <0.0001* |
| Urine ACR (mg/g) | 4.5 ± 18.9 | 10.8 ± 54.3 | 34.1 ± 74.2 | 247.4 ± 325.2 | <0.0001* | <0.0001* |
| Comorbidity | | | | | | |
| Hypertension (%) | 18/ 66 (27.3%) | 84/ 303 (27.7%) | 44/ 119 (37.0%) | 2/ 9 (22.2%) | 0.2611 | 0.1822 |
| Hyperlipidemia (%) | 29/ 66 (43.9%) | 142/ 303 (46.9%) | 45/ 119 (37.8%) | 3/ 9 (33.3%) | 0.3518 | 0.1977 |
| Diabetes mellitus (%) | 14/ 66 (21.2%) | 58/ 303 (19.1%) | 37/ 119 (31.1%) | 3/ 9 (33.3%) | 0.0536 | 0.0326 |
| Blood pressure | | | | | | |
| Systolic (mmHg) | 135.1 ± 17.9 | 132.6 ± 18.4 | 135.5 ± 19.8 | 133.8 ± 12.8 | 0.5505 | 0.6271 |
| Diastolic (mmHg) | 76.7 ± 10.0 | 76.3 ± 10.4 | 74.8 ± 11.7 | 73.7 ± 16.5 | 0.6127 | 0.2032 |
| MAP (mmHg) | 96.2 ± 11.5 | 95.0 ± 12.0 | 95.1 ± 13.2 | 93.7 ± 14.5 | 0.9296 | 0.6174 |
| Serum laboratory studies | | | | | | |
| Fasting glucose (mg/dL) | 109.9 ± 34.0 | 108.5 ± 23.9 | 113.5 ± 36.9 | 105.4 ± 19.3 | 0.4485 | 0.4189 |
| HbA1c (%) | 6.1 ± 1.2 | 6.1 ± 0.8 | 6.3 ± 0.9 | 6.5 ± 0.8 | 0.0430 | 0.0224 |
| Total cholesterol (mg/dL) | 186.2 ± 34.3 | 188.2 ± 34.8 | 180.9 ± 38.4 | 175.7 ± 32.7 | 0.2245 | 0.1198 |
| LDL-C (mg/dL) | 116.1 ± 33.3 | 118.7 ± 31.3 | 115.6 ± 32.3 | 117.6 ± 14.5 | 0.8215 | 0.7705 |
| HDL-C (mg/dL) | 57.2 ± 15.4 | 53.8 ± 15.3 | 48.8 ± 14.0 | 43.7 ± 10.9 | 0.0003* | <0.0001* |
| Triglycerides (mg/dL) | 115.9 ± 74.6 | 120.1 ± 71.3 | 121.5 ± 65.0 | 105.9 ± 56.2 | 0.8900 | 0.8316 |

Abbreviations: ACR, albumin-to-creatinine ratio; CKD, chronic kidney disease; eGFR, estimated glomerular filtration rate; HDL-C, high-density lipoprotein cholesterol; LDL-C, low-density lipoprotein cholesterol; MAP, mean arterial pressure; US dollars.

*Statistical significance with Bonferroni correction, P < 0.0125.

pressures of the four CKD groups were within normal limits, and no significant difference was observed in the mean arterial pressure. In serum laboratory testing, no significant difference was observed in the levels of total cholesterol, LDL-C, triglycerides, and fasting glucose. The levels of HbA1c and HDL-C were not homogenous among the four groups of CKD; however, the mean HbA1c and HDL-C levels were within normal limits.

## Mini-Mental State Examination (MMSE) and life quality assessment

Our assessment of cognitive function with MMSE scores and quality of life for all participants is shown in Table 2. Compared to groups 1 and 2, groups 3 and 4 showed lower scores for each domain of the MMSE. On average, the global MMSE scores for all participants in groups 1, 2, 3, and 4 were 27.4 ± 2.4, 27.4 ± 2.8, 25.3 ± 4.7, and 24.9 ± 3.4, respectively ($P < 0.0001$ and trend $P < 0.0001$). The scores for the domains of calculation ($P < 0.0001$ and trend $P < 0.0001$), memory recall ($P = 0.0018$ and trend $P = 0.0012$), and complex commands ($P = 0.0105$ and trend $P = 0.0047$) differed significantly among the four groups. While the scores for the quality of life questionnaire, including dimensions of VT, SF, RE, and MH, did not differ significantly among the four groups, the RP score was significantly impaired in the

**Table 2. Mini-Mental State examination and quality of life assessment of all participants ($N = 497$).**

| CKD Staging | Stage 1 ($N = 66$) | Stage 2 ($N = 303$) | Stage 3a, 3b ($N = 119$) | Stage 4–5 ($N = 9$) | P value | P trend |
|---|---|---|---|---|---|---|
| Global MMSE (30 points) | 27.4 ± 2.4 | 27.4 ± 2.8 | 25.3 ± 4.7 | 24.9 ± 3.4 | <0.0001* | <0.0001* |
| Orientation to time (5 points) | 4.9 ± 0.4 | 4.9 ± 0.5 | 4.6 ± 0.9 | 4.3 ± 0.9 | <0.0001* | <0.0001* |
| Orientation to place (5 points) | 4.9 ± 0.4 | 4.9 ± 0.3 | 4.8 ± 0.7 | 5.0 ± 0.0 | 0.1011 | 0.1627 |
| Registration (3 points) | 2.9 ± 0.4 | 3.0 ± 0.2 | 2.9 ± 0.4 | 2.8 ± 0.4 | 0.1227 | 0.0883 |
| Calculation (5 points) | 4.1 ± 1.3 | 4.0 ± 1.4 | 3.3 ± 1.6 | 2.9 ± 1.7 | <0.0001* | <0.0001* |
| Memory recall (3 points) | 2.5 ± 0.9 | 2.4 ± 0.8 | 2.1 ± 1.0 | 2.2 ± 1.0 | 0.0018* | 0.0012* |
| Language (2 points) | 2.0 ± 0.1 | 2.0 ± 0.1 | 1.9 ± 0.3 | 2.0 ± 0.0 | 0.3403 | 0.2250 |
| Repetition (1 point) | 0.9 ± 0.3 | 0.9 ± 0.3 | 0.8 ± 0.4 | 0.8 ± 0.4 | 0.2929 | 0.0900 |
| Complex commands (6 points) | 5.3 ± 1.0 | 5.3 ± 1.1 | 4.9 ± 1.4 | 4.9 ± 0.9 | 0.0105* | 0.0047* |
| Quality of life assessment | | | | | | |
| Role-physical (RP) | 51.2 ± 9.9 | 50.5 ± 9.6 | 48.4 ± 10.6 | 43.0 ± 12.0 | 0.0288* | 0.0074* |
| Vitality (VT) | 51.6 ± 9.8 | 50.1 ± 9.6 | 49.1 ± 11.0 | 46.0 ± 11.1 | 0.2707 | 0.0569 |
| Social functioning (SF) | 49.3 ± 11.4 | 50.8 ± 8.7 | 48.8 ± 11.2 | 44.1 ± 19.2 | 0.0664 | 0.1651 |
| Role-emotional (RE) | 49.3 ± 11.0 | 50.3 ± 9.7 | 49.5 ± 10.3 | 48.8 ± 11.4 | 0.7925 | 0.8429 |
| Mental health (MH) | 50.2 ± 9.4 | 50.1 ± 10.3 | 49.5 ± 9.7 | 50.0 ± 10.5 | 0.9579 | 0.6541 |

CKD, chronic kidney disease; MMSE, Mini-Mental State Examination.

*Statistical significance with Bonferroni correction $P < 0.0125$.

late stages of CKD. The stratification analysis by sex is shown in S1 Table. Similar to the analysis for all participants, the scores of global MMSE, and the domains of calculation, memory recall, and complex commands were significantly impaired in stages 3a and 3b for both female and male participants. In addition, scores of domains of orientation to place, registration, and memory were significantly impaired in stages 3a and 3b for females.

## Association between cognitive impairment and renal function decline with generalized linear mixed models (GLMMs)

The association between impaired eGFR and each dimension of the MMSE and quality of life questionnaire was examined in adjusted GLMMs (Table 3). The relationships among eGFR, age, and sex for global MMSE (Fig 1) and each domain of the MMSE (Fig 2) were plotted. Lowering eGFR per 30 mL/min/1.73 m$^2$ was significantly associated with impaired global MMSE score (β = −0.963, standard error [SE] = 0.239, adjusted $P < 0.0001$), domains of orientation to time (β = −0.161, SE = 0.456, adjusted $P = 0.0004$), and of calculation (β = −0.397, SE = 0.109, adjusted $P = 0.0003$). For quality of life assessment, lowering eGFR per 30 mL/min/1.73 m$^2$ was associated with impaired scores of RP (β = −2.387, SE = 0.687, adjusted $P = 0.0006$), but no obvious changes in the dimensions of VT, SF, RE, and MH (Fig 3). The estimated regression parameters **β** of eGFR in unadjusted and adjusted GLMMs (Models 1–4) are shown in Table 4.

## Validation study with partial least squares procedures

The Spearman's correlation coefficients for all explanatory variables are listed in S2 Table). While all items in the MMSE were very weakly correlated, except for the weak association between calculation and complex commands (ρ = 0.374), all dimensions in SF-12 were weakly correlated, except for the moderate association between vitality and mental health (ρ = 0.566). The results of the validation study are listed in Table 5. Using PLS regression with PCR, we extracted two principal component factors from the dimensions of the MMSE and quality of

**Table 3. Generalized linear mixed model estimates of Mini-Mental State examination and quality of life (adjusted by age and sex).**

| Scores of each items (*N* = 497) | eGFR (per 30 mL/min/1.73 m² lower†) | | | Age (per years increase) | | | Sex (female versus male) | | | Age × sex | | |
|---|---|---|---|---|---|---|---|---|---|---|---|---|
| | β | S.E. | *P* value | β | S.E. | *P* value | β | S.E. | *P* value | β | S.E. | *P* value |
| Global MMSE | -0.963 | 0.239 | <0.0001* | -0.067 | 0.030 | 0.0280 | 10.475 | 3.483 | 0.0028* | -0.169 | 0.048 | 0.0005* |
| Orientation to time | -0.161 | 0.456 | 0.0004* | -0.001 | 0.001 | 0.1435 | -0.931 | 0.665 | 0.1619 | -0.015 | 0.001 | 0.1086 |
| Orientation to place | -0.014 | 0.030 | 0.6299 | 0.002 | 0.004 | 0.6125 | 1.582 | 0.437 | 0.0003* | -0.024 | 0.006 | <0.0001* |
| Registration | -0.043 | 0.024 | 0.0685 | 0.000 | 0.002 | 0.9229 | 0.338 | 0.343 | 0.3242 | -0.006 | 0.005 | 0.2383 |
| Calculation | -0.397 | 0.109 | 0.0003* | -0.029 | 0.014 | 0.0346 | 2.133 | 1.582 | 0.1781 | -0.036 | 0.022 | 0.0964 |
| Memory recall | -0.106 | 0.063 | 0.0929 | -0.020 | 0.008 | 0.0133 | 1.949 | 0.918 | 0.0343 | -0.031 | 0.013 | 0.0154 |
| Language | -0.007 | 0.015 | 0.6566 | -0.001 | 0.002 | 0.6425 | 0.205 | 0.215 | 0.3419 | -0.003 | 0.003 | 0.2882 |
| Repetition | -0.030 | 0.025 | 0.2317 | -0.002 | 0.003 | 0.6064 | 0.179 | 0.366 | 0.6252 | -0.002 | 0.005 | 0.6263 |
| Complex commands | -0.205 | 0.082 | 0.0132 | -0.001 | 0.010 | 0.3897 | 3.158 | 1.202 | 0.0088 | -0.051 | 0.017 | 0.0022* |
| Quality of life | | | | | | | | | | | | |
| Role-physical (RP) | -2.332 | 0.756 | 0.0023* | -0.141 | 0.094 | 0.1333 | -2.976 | 10.993 | 0.7867 | 0.006 | 0.152 | 0.9694 |
| Vitality (VT) | -1.815 | 0.758 | 0.0170 | 0.012 | 0.095 | 0.9000 | 6.217 | 11.095 | 0.5755 | -0.125 | 0.154 | 0.4177 |
| Social functioning (SF) | -1.438 | 0.764 | 0.0603 | 0.057 | 0.096 | 0.5522 | 2.904 | 11.195 | 0.7954 | -0.036 | 0.155 | 0.8183 |
| Role-emotional (RE) | -0.680 | 0.761 | 0.3716 | -0.058 | 0.095 | 0.5433 | 5.376 | 11.151 | 0.0227 | -0.104 | 0.155 | 0.5024 |
| Mental health (MH) | -0.514 | 0.766 | 0.5028 | 0.012 | 0.096 | 0.8992 | -1.449 | 11.241 | 0.8975 | -0.002 | 0.156 | 0.9877 |

eGFR, estimated glomerular filtration rate; MMSE, Mini-Mental State Examination. S.E., standard error.

*Statistical significance with Bonferroni correction, *P* < 0.0125.

† A lower eGFR per 30 mL/min/1.73 m² indicates much worsened renal function in the early and moderate stages of chronic kidney disease.

life questionnaire. Factors 1 and 2 accounted for 2.3% and 0.5% of the total variation in eGFR (the dependent variable), respectively. Compared with Factor 1, Factor 2 was very weak in explaining the variation in eGFR. Factors 1 and 2 accounted for 23.5% and 18.2% of the total variation in the PCR model effects, respectively. For Factor 1, the independent variables orientation to time (0.337), orientation to place (0.397), registration (0.324), calculation (0.318), and complex commands (0.354) had factor loadings of >|0.3| and were significant in explaining the variation in eGFR. In contrast, for factor 2, the independent variables RP (0.397), VT (0.411), SF (0.339), RE (0.394), and MH (0.409) had high factor loadings, and they were very weak in explaining the variation in eGFR. These results are compatible with our findings from GLMMs.

## Discussion

In this study, we investigated multiple domains of cognitive function and quality of life among community-based healthy older adults with early CKD. Participants with early CKD stages 3a and 3b had poor orientation to time, calculation, and executive function, and lower scores for physical status in the SF-12. Memory function showed no overt impairment in early CKD. The study results expand the previous findings [27, 33] that executive function deficits are evident earlier in the CKD illness at stage 3, and patients with late CKD stages 4–5 with poorer orientation in the older Han Chinese population. We found impaired executive function in the early stage of CKD (stages 3a and 3b), and global cognition impairment was accompanied by life dissatisfaction with RP and VT in early CKD stages 3a and 3b. This suggests that early intervention for cognitive assessment should be conducted in CKD stages 3a and 3b in older Han Chinese adults.

The speculative causes for the association between cognitive impairment and renal function deterioration were grouped into four categories: demography, clinical characteristics, dialysis

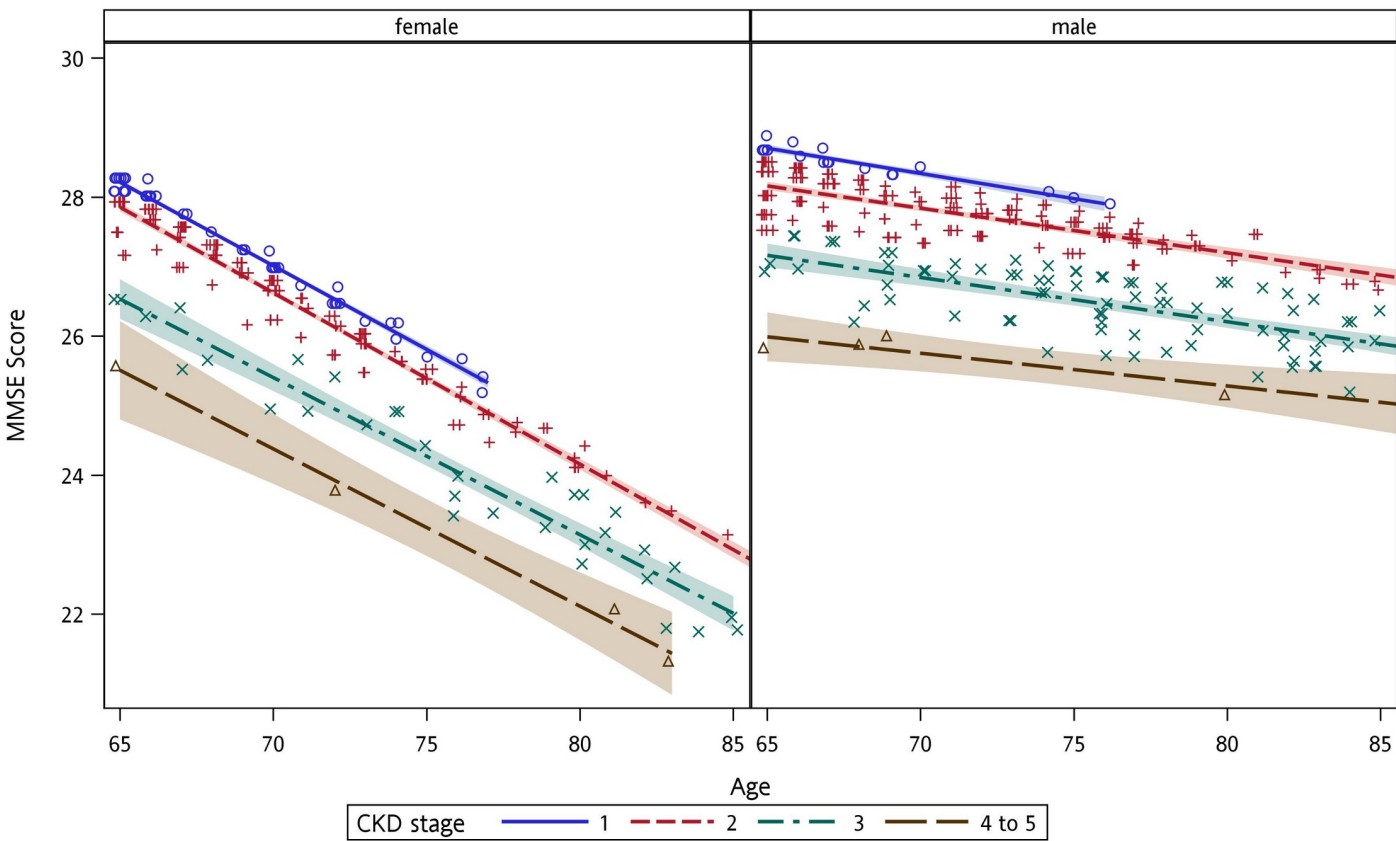

**Fig 1. Global Mini-Mental State Examination (MMSE) score by age, sex, and chronic kidney disease (CKD) stages.**

procedures, and vascular factors [15–17, 69–71]. First, demographic contribution should not be a single factor to explain cognitive impairment in our participants with early stage of CKD. After adjustment for age and sex in models 2–4, the GLMMs showed that a lower eGFR was independently associated with cognitive impairment. Second, clinical factors did not explain the differences in cognitive impairment among the CKD groups. Third, none of the participants with CKD stages 3a and 3b received hemodialysis. This confirms the rebuttal association between cognitive impairment and hemodialysis [72]. Fourth, we considered that vascular factors also contributed to the causes of cognitive impairment in our early CKD participants (stages 3a and 3b). Vascular factors have been reported to be implicated in frontal-temporal lobe injury in early CKD, including interference with orientation and executive dysfunction [73, 74]. Unsurprisingly, therefore, our analysis supported the presence of cognitive impairment in the early to moderate stages of CKD in Han Chinese older adults. Our results are consistent with an earlier systematic review [29] that suggested that cognitive impairment exists across all stages of CKD, independent of the aging effect.

For quality of life assessment, our participants showed no obvious deficits in vitality, social functioning, emotional role, and mental health, except for physical role function. Our analysis was mostly consistent with a recent study [42] employing the EQ-VAS, which showed impaired self-perceived quality of life in patients at CKD stages 3a, 3b, and 4 before and after adjusting for sociodemographic factors. However, the EQ-VAS is a multifactorial scale, which

## Each Category of MMSE Scores by Age, sex, and CKD Stages

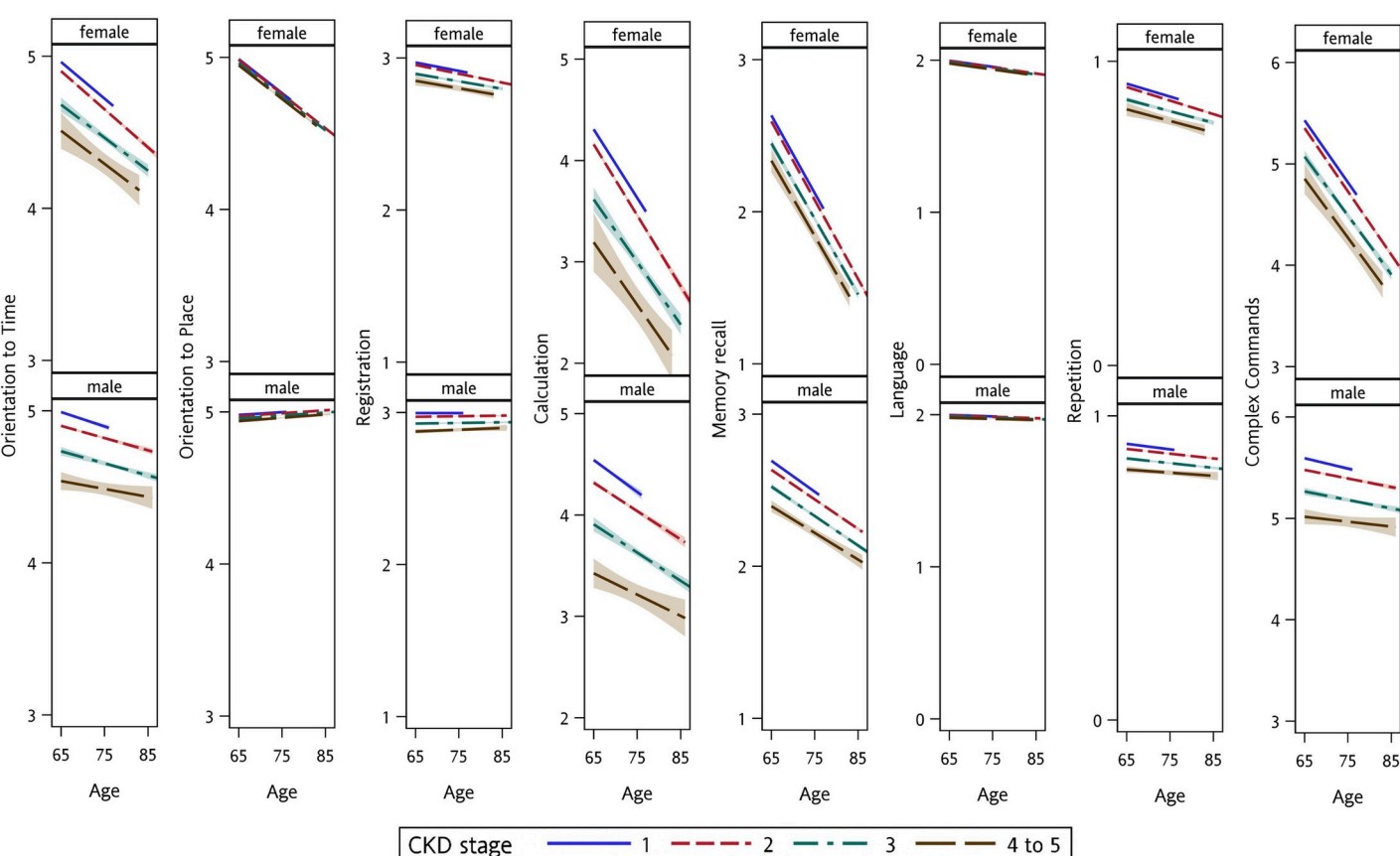

**Fig 2. Each category of Mini-Mental State Examination (MMSE) score by age, sex, and chronic kidney disease (CKD) stages.**

was scored by asking the participants to indicate their health state in a range from 0 (worst) to 100 (best). Additionally, our results are similar to those of another study [40] using the SF-8, which showed worse physical component scores before adjustment for age and medical comorbidities in all stages of CKD. On the other hand, our assessment of quality of life is incompatible with a study [41] assessing quality of life using the CASP-19 scale. We considered that the dimensions of quality of life measured by the SF-12 and CASP-19 scales are different. The CASP-19 is much focused on self-completion, pleasure, and control [75].

Compared to previous studies, our study has the following key characteristics: First, previous analysis found unclear which is the best method for assessing eGFR in elderly and cognitively impaired adults [32, 60]. Since we enrolled a monoethnic Chinese population, we could manage this problem by employing the BIS equation. Our conclusion is consistent with previous studies [29, 32, 33] using the MDRD and CKD-EPI equations in different ethnic and age groups. Second, our analysis included covariates of demography, sociodemographic factors, and medical comorbidities in terms of laboratory tests. Third, we performed the analysis in both genders, unlike some studies with only men [76] and women [35, 77]. An earlier study in Taiwan [35] included only midlife women in Kinmen, a group of islands off the southeastern coast of mainland China, governed as a county in Taiwan. Fourth, our nationwide sampling should have a representative population of Han Chinese. We performed a comprehensive door-to-door survey conducted by trained interviewers and laboratory examinations. Fifth, we

## Quality of Life Questionnaires by Age, Sex, and CKD Stages

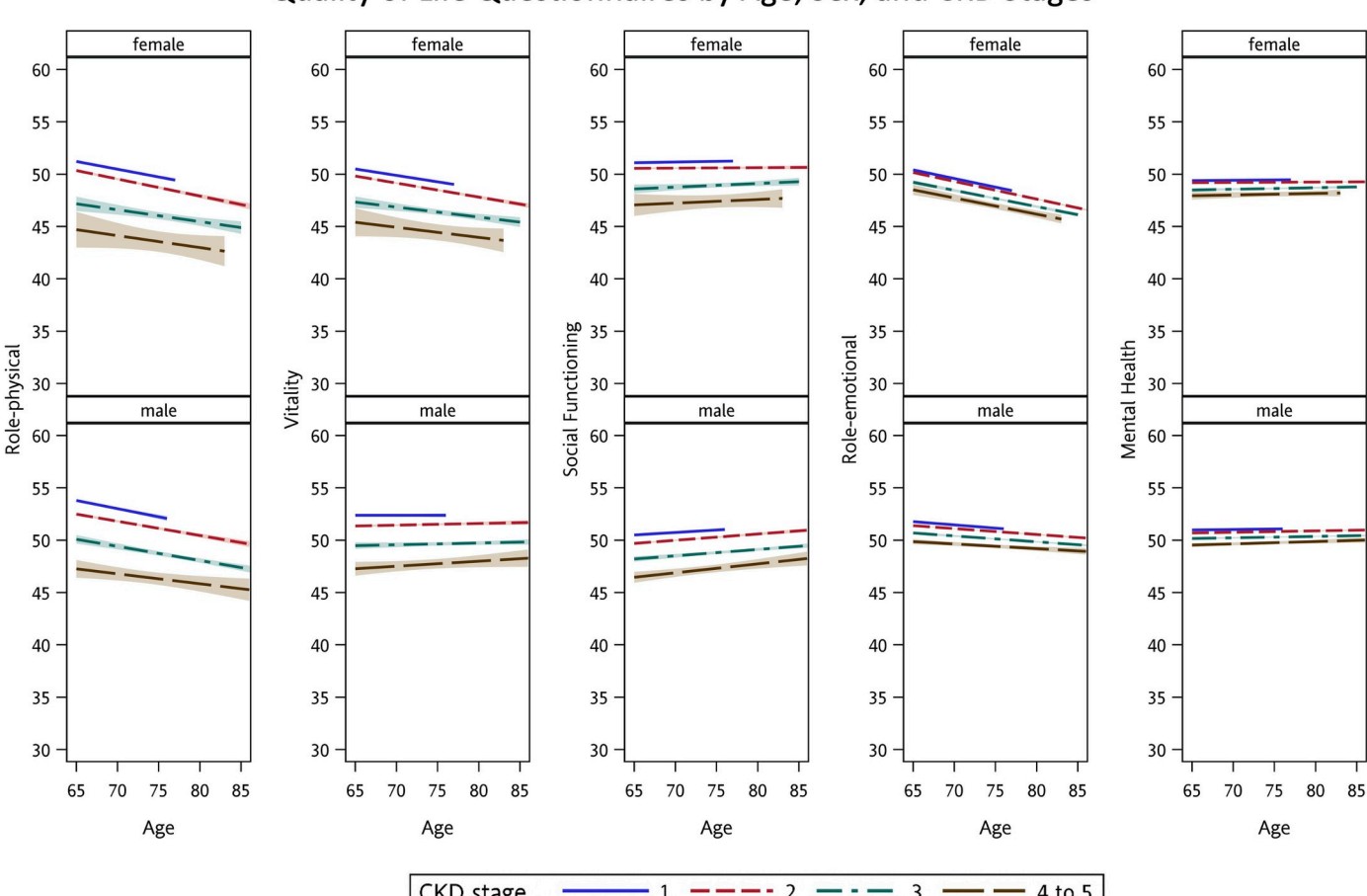

**Fig 3. Quality of life questionnaires by age, sex, and chronic kidney disease (CKD) stages.**

validated the study results using the unconventional method of partial least squares regressions with principal components to investigate the factor loadings on the response variable of eGFR. Factor loadings on orientation, calculation, and complex domains were all > 0.3, which is consistent with the analysis of GLMMs. Sixth, cognition and quality of life were assessed in our study.

The study has some limitations. First, our study design involved a cross-sectional design, which cannot be used to infer the cause-effect relationship. Since this study was part of a large-scale nationwide nutrition survey, more advanced biomarkers or imaging studies were unavailable for our data collection. Second, despite being the most widely used instrument in the extant literature, the MMSE is known to be affected by ceiling and floor effects in different languages and populations [29, 78, 79]. While the ceiling effect indicates that the performance of MMSE, independent of cognitive state, is favorably affected by highly educated level, the floor effect suggests that performance is adversely affected by poor education [78]. In addition, the MMSE is also known to be less sensitive to mild changes in cognition, and therefore may have reduced sensitivity for early CKD stages [29]. These could limit the generalizability and comparability of similar studies. Some studies [80, 81] indicated that the Montreal Cognitive Assessment (MoCA) showed better utility as a tool for assessing patients with CKD. Third, we did not assess severe cardiovascular diseases or other neurological disorders in our study.

**Table 4. Comparison of different models for the relationship among eGFR estimates, MMSE, and quality of life.**

| Models (N = 497) | Model 1 (unadjusted) | | | Model 2 (adjusted for age, sex, and age × sex) | | | Model 3 (adjusted for age, sex, age × sex, education years) | | | Model 4 (adjusted for age, sex, age × sex, education years, HbA1c, and HDL-C) | | |
|---|---|---|---|---|---|---|---|---|---|---|---|---|
| eGFR (per 30 mL/min/1.73 m² lower†) | β | S.E. | *P* value | β | S.E. | *P* value | β | S.E. | *P* value | β | S.E. | *P* value |
| **Global MMSE** | -1.180 | 0.230 | <0.0001* | -0.963 | 0.239 | <0.0001* | -0.922 | 0.228 | <0.0001* | -0.807 | 0.235 | 0.0007* |
| **Orientation to time** | -0.192 | 0.042 | <0.0001* | -0.161 | 0.456 | 0.0004* | -0.159 | 0.046 | 0.0005* | -0.155 | 0.047 | 0.0011* |
| **Orientation to place** | -0.017 | 0.028 | 0.5433 | -0.014 | 0.030 | 0.6299 | -0.013 | 0.030 | 0.6721 | -0.001 | 0.031 | 0.7835 |
| **Registration** | -0.040 | 0.021 | 0.0645 | -0.043 | 0.024 | 0.0685 | -0.042 | 0.023 | 0.0750 | -0.038 | 0.024 | 0.1152 |
| **Calculation** | -0.475 | 0.101 | <0.0001* | -0.397 | 0.109 | 0.0003* | -0.383 | 0.106 | 0.0003* | -0.338 | 0.109 | 0.0020* |
| **Memory recall** | -0.178 | 0.059 | 0.0027* | -0.106 | 0.063 | 0.0929 | -0.101 | 0.062 | 0.1068 | -0.084 | 0.064 | 0.1953 |
| **Language** | -0.011 | 0.013 | 0.4297 | -0.007 | 0.015 | 0.6566 | -0.001 | 0.015 | 0.6675 | -0.001 | 0.015 | 0.8341 |
| **Repetition** | -0.040 | 0.023 | 0.0800 | -0.030 | 0.025 | 0.2317 | -0.030 | 0.025 | 0.2387 | -0.024 | 0.026 | 0.3540 |
| **Complex commands** | -0.228 | 0.077 | 0.0035* | -0.205 | 0.082 | 0.0132 | -0.188 | 0.077 | 0.0150 | -0.156 | 0.079 | 0.0494 |
| **Quality of life** | | | | | | | | | | | | |
| **Role-physical (RP)** | -2.387 | 0.687 | 0.0006* | -2.332 | 0.756 | 0.0023* | -2.301 | 0.757 | 0.0024* | -2.219 | 0.779 | 0.0046* |
| **Vitality (VT)** | -1.480 | 0.691 | 0.0327 | -1.815 | 0.758 | 0.0170 | -1.784 | 0.757 | 0.0189 | -1.929 | 0.777 | 0.0134 |
| **Social functioning (SF)** | -1.339 | 0.690 | 0.0530 | -1.438 | 0.764 | 0.0603 | -1.421 | 0.764 | 0.0635 | -1.403 | 0.781 | 0.0730 |
| **Role-emotional (RE)** | -0.686 | 0.692 | 0.3217 | -0.680 | 0.761 | 0.3716 | -0.654 | 0.760 | 0.3905 | -0.809 | 0.784 | 0.3023 |
| **Mental health (MH)** | -0.204 | 0.695 | 0.7693 | -0.514 | 0.766 | 0.5028 | -0.493 | 0.766 | 0.5199 | -0.896 | 0.773 | 0.2472 |

eGFR, estimated glomerular filtration rate; MMSE, Mini-Mental State Examination. S.E., standard error.

*Statistical significance with Bonferroni correction, $P < 0.0125$.

†A lower eGFR per 30 mL/min/1.73 m² indicates much worsened renal function in the early and moderate stages of chronic kidney disease.

**Table 5. Validation with partial least squares (PLS) procedures.**

| Methods of PLS Procedures | Principle component regression | |
|---|---|---|
| | **Factor 1** | **Factor 2** |
| Percent Variation Accounted for by Principal Components (%) | | |
| Model effects | 23.475 | 18.171 |
| Dependent variables of eGFR | 2.309 | 0.478 |
| Model effect loadings | | |
| Age | -0.154 | -0.000 |
| Orientation to time | 0.337* | -0.201 |
| Orientation to place | 0.397* | -0.241 |
| Registration | 0.324* | -0.182 |
| Calculation | 0.318* | -0.047 |
| Memory recall | 0.261 | -0.066 |
| Language | 0.296 | -0.282 |
| Repetition | 0.165 | 0.031 |
| Complex commands | 0.354* | -0.136 |
| Role-physical (RP) | 0.207 | 0.397* |
| Vitality (VT) | 0.224 | 0.411* |
| Social functioning (SF) | 0.150 | 0.339* |
| Role-emotional (RE) | 0.188 | 0.394* |
| Mental health (MH) | 0.200 | 0.409* |

*Variables with a factor loading of > |0.3| were considered significant.

Since the NAHSIT 2013–2016 primarily focused on healthy older adults in adults, only comorbidities of hypertension, hyperlipidemia, and diabetes mellitus were assessed in our study.

Our study confirmed that CKD with cognitive impairment (especially the domains of orientation, calculation, and complex commands) is present in the early to moderate illness stages. A non-invasive behavior intervention by cognitive remediation (CR) is proposed to help promote neuroplastic change and enhance cognitive performance in patients with CKD [82]. Different CR approaches [83] can be tailored to specific cognitive deficits in patients with CKD. On the other hand, pharmacological intervention with erythropoietin, a hormone used for the treatment of renal anemia, showed possible neuroprotective effects in animals [84, 85] and human studies [86, 87]. Other agents, such as cholinesterase inhibitors, have been used in patients with Alzheimer's disease [88] and vascular dementia [89], but not specifically for CKD patients. However, to date, there has been a shortage of behavioral interventions and pharmacological research on cognitive improvement in CKD patients [82].

## Conclusions

In conclusion, this nationwide, community-based, cross-sectional study in healthy older adults confirmed that early CKD (stages 3a and 3b) is associated with cognitive decline in the global MMSE and domains of orientation to time, calculation, and complex commands for the elderly Han Chinese population. More attention should be paid to healthy older adults with early stage CKD with physical role function deficits in quality of life. A prospective cohort study with a longer follow-up period might provide more evidence regarding the cause-effect relationship and thus enable a clearer understanding of the nature of the relationship. Strategies to manage cognitive impairment and improve quality of life can be implemented accordingly.

## Supporting information

**S1 Table. Mini-Mental State examination and quality of life assessment stratified by sex.**
(DOCX)

**S2 Table. Spearman correlation coefficients among all items of the Mini-Mental State examination and SF-12.**
(DOCX)

## Author Contributions

**Conceptualization:** Sheng-Feng Lin, Wen-Harn Pan.

**Data curation:** Yen-Chun Fan, Tzu-Tung Kuo.

**Formal analysis:** Sheng-Feng Lin.

**Funding acquisition:** Chyi-Huey Bai.

**Investigation:** Chyi-Huey Bai.

**Writing – original draft:** Sheng-Feng Lin.

**Writing – review & editing:** Sheng-Feng Lin, Chyi-Huey Bai.

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
