## [Decision Letter · Decision Letter 0]

17 Aug 2021

PONE-D-20-31145

Quality of Life and Cognitive Assessment in Healthy Older Asian People with Early Chronic Kidney Disease: the NAHSIT 2013–2016 and Validation Study

PLOS ONE

Dear Dr. Bai,

Thank you for submitting your manuscript to PLOS ONE. After careful consideration, we feel that it has merit but does not fully meet PLOS ONE’s publication criteria as it currently stands. Therefore, we invite you to submit a revised version of the manuscript that addresses the points raised during the review process.

The manuscript has been evaluated by two reviewers, and their comments are available below.

The reviewers have raised a number of concerns regarding the manuscript’s clarity and organization. They specifically request greater support and clarification of the manuscript’s theoretical framework and details of the methodology. They also note greater depth of discussion and organization in the Introduction and Discussion, as well as a thorough revision for language and grammar.

Could you please carefully revise the manuscript to address all comments raised?

We look forward to receiving your revised manuscript.

Kind regards,

Avanti Dey, PhD

Staff Editor

PLOS ONE

Journal Requirements:

4. We note you have included a table to which you do not refer in the text of your manuscript. Please ensure that you refer to Table 5 in your text; if accepted, production will need this reference to link the reader to the Table.

Reviewers' comments:

Reviewer's Responses to Questions

**Comments to the Author**

1. Is the manuscript technically sound, and do the data support the conclusions?

Reviewer #1: Partly

Reviewer #2: Yes

2. Has the statistical analysis been performed appropriately and rigorously? 

Reviewer #1: No

Reviewer #2: Yes

3. Have the authors made all data underlying the findings in their manuscript fully available?

Reviewer #1: No

Reviewer #2: No

4. Is the manuscript presented in an intelligible fashion and written in standard English?

Reviewer #1: No

Reviewer #2: No

5. Review Comments to the Author

Reviewer #1: “Quality of Life and cognitive assessment in healthy older Asian people with early chronic kidney disease: the NAHSIT 2013-2016 and validation study.”: Investigators intended to study the association between kidney function and cognition, as well as quality of life. The study is based on data from the NAHSIT dataset. Authors concluded that CKD stages 3 (a and b) is associated with cognitive decline in multiple domains, and that these patients have physical complaints.

The paper requires editing for English, for clarification of meaning.

Abstract

Purpose of the study is not stated.

Need to mention study is Taiwan-based.

In methods, need to mention this is data from NAHSIT 2013-2016.

Also it says “…categorized into three CKD groups”… but they spell out 4 groups.

Authors refer to those in stages 3 as “early CKD”; those would be patients with “moderate” disease.

In conclusion, “multiple cognitive decline”, “More attention for cognition should be paid for healthy older adults perceiving dissatisfaction on physical status”: unclear what these mean.

Introduction

-1st paragraph (prg): authors refer to USRDS annual report but don’t cite it.

-The introduction is inadequate; the argument for the study is weak.

-The specific aim/hypothesis need to be rewritten (in line with design) for clarity.

-Literature on cognition and CKD is not substantiated, despite the availability of important existing literature (such as with various tests of cognition performed on the Chronic Renal Insufficiency Cohort).

-Authors do not make the case for studying quality of life (QOL).

-Unclear why only “physical disability” and “caregiver burden” are highlighted.

Methods

-“systemic sampling”: do authors mean “systematic”?

-Authors need to clarify if this is a secondary analysis of data (2013-2016 of NAHSIT); otherwise, are they reporting primary data collection on MMSE/SF-12 as an add-on (to NAHSIT) conducted in those years?? This needs to be clear. What was the main aim of constructing the primary dataset?

-Authors report this to be a “cross-sectional” study; need to clarify if all participant data were collected in only one visit.

-Data collection: clarify that this was for the primary study. Unclear how lab results on glucose and lipids, methodology on BP measurement are relevant in the context of this work. It would be helpful instead to elaborate on sociodemographic data for the current study.

-There is very limited description of measurement tools (MMSE, SF-12) pertinent to main study variables; additional details are needed such as SF-12 scoring and interpretation, applicability in Asian population, other.

-The use of Berlin Initiative Study equation for GFR estimation is not mainstream; authors did not elaborate on its applicability in Asian populations. Authors haven’t explained why those in stages 4 and 5 are grouped into one category. In labeling patients in stages 1 and 2 of CKD, how the diagnosis was made in addition to eGFR needs to be clarified.

-Statistical analysis: Elaboration is needed on the use of GLMM in this cross-sectional study. Authors need to clarify “covariates” used in the modeling, instead of referring to them as independent variables. The validation approach using principal component regression (factors 1 and 2, with dependent variable of eGFR) is unclear.

Results

-How was the 30-point change in eGFR coined in the study?

-In table 4, why does model 4 include HbA1C and HDL?

-Validation study and results depicted in figures (depicting outcomes by age, sex)… are not in line with study purpose (there are no Specific Aims to support the conduct of those analyses).

Discussion and Conclusion:

-The word “begin” would relate to longitudinal studies; its use should be reconsidered in this paper.

-Discussion/conclusion will require revision based on revisions in previous sections.

-Unclear what is meant by “feeling physical disturbance” in conclusion.

Reviewer #2: The authors report on a large nationwide study of cognitive impairment and quality of life in CKD patients 65 years and above. There are a couple of areas that can be improved, particularly in terms of study rationale, terminology and phrasing of interpretations, which currently reduce my enthusiasm for this manuscript.

Line 31-32: Merge CKD stages 1 and 2 to match the statement that 3 groups were derived. At present it reads as 4 separate groups.

The Introduction itself is short and lacking on details that contribute to a well-supported study rationale. In particular, the authors have not provided any evidence from previously published papers showing that cognitive impairment is related to eGFR levels (see Etgen T et al Chronic kidney disease and cognitive impairment: a systematic review and meta-analysis. Am J Nephrol. 2012;35(5):474–82), which would support the notion that cognitive impairment would be present in early as well as late stage CKD. In fact, this is evidenced in the Berger et al. 2016 systematic review and meta-analysis which the authors cite in the Discussion. Furthermore, in another systematic review (Brodski, J. et al. (2019). A Systematic Review of Cognitive Impairments Associated With Kidney Failure in Adults Before Natural Age-Related Changes. Journal of the International Neuropsychological Society, 25(1), 101-114.), it has been shown that the cognitive impairments found in relation to CKD are independent of age-related cognitive decline, and incrementally increase as the stages of CKD advance. At the very least, these key reviews/meta-analyses should be cited to support the argument for the authors investigating cognition in early-stage CKD, even if within and despite an older cohort.

The Introduction is also lacking in any background information to support an investigation of quality of life within a CKD sample. Please add this information.

Based on the existing information in the literature, once added to the Introduction, the authors should revise the hypotheses to be a bit more specific. I would also recommended a statement of aims to cover the components of the proposed analyses.

The authors are further requested to clarify and provide a rationale for the validation study within the Introduction section. At present, it appears abruptly.

Was information about co-morbid conditions (e.g. diabetes, cardiovascular disease, neurological disorders/conditions)? There is a sizeable literature showing cognitive impairments associated with these conditions, which need to be considered for the current dataset. If this information was not collected, it is a major limitation of the study and needs to be mentioned in text.

Given the number of analyses conducted, correction for multiple comparisons (e.g. Bonferroni correction) in the group comparisons is required to minimise Type 1 error.

Participant group characteristics could be reported in a table for greater clarity and ease of comprehension.

It is not appropriate to say that MMSE scores decreased between stages as the participants were not followed up longitudinally. As what has been conducted are just group comparisons, the differences should be described as performance differences between groups of patients at the different stages. The use of ‘decreasing’ is misleading in this context of reporting group comparison results. In addition, within the adjusted GLMMs, it should also be phrased similarly (e.g. lower eGFR is associated with reduced QOL). Please amend throughout.

In line 205, please refrain from using terminology such as ‘thoroughly investigated’ when many cognitive domains were not assessed in this study.

In lines 210-211, it is not accurate to say that “patients of late CKD stages 4–5 with poorer orientation, executive function begin in the early stage of CKD (stages 3a and 3b)” as the authors did not longitudinally follow a group of patients as their CKD progressed. I believe the authors are suggesting that executive function deficits are present earlier in the illness at Stage 3, so it would be more accurate to just say that executive function deficits are evident from Stage 3 CKD.

In line 220, increasing age and female sex cannot promote cognitive impairment, rather they can be associated with greater cognitive impairment. Please rephrase.

In line 224, it would be better to say that clinical factors do not really explain the differences in cognitive impairment between the groups.

Line 230-232: It is not intuitive how the authors arrived at the conclusion that their analysis “analysis supported the association between early renal function impairment and cognitive impairment should not be the confounding effect of aging only” based on the statement preceding it. Please clarify. In addition, such a conclusion has previously been reached in the literature (Brodski et al, 2019). Relatedly, the authors should better integrate and situate their findings in relation to the existing literature.

Line 234-242: It is unclear how this paragraph fits within the narrative.

Regarding limitations, the authors should also discuss the limits of using the MMSE for assessing cognitive function within CKD (see Brodski et al, 2019). Also see my earlier point about premorbid conditions.

The authors should also briefly discuss the utility of cognitive remediation to address cognitive deficits, particularly in early CKD (see Tan, E. J. et al (2019). Considering the utility of cognitive remediation therapy in chronic kidney disease. Clinical and Experimental Nephrology.) for some points. This would help better contextualise the current findings.

Please review the manuscript for grammatical errors throughout.

6. PLOS authors have the option to publish the peer review history of their article (what does this mean?). If published, this will include your full peer review and any attached files.

Reviewer #1: No

Reviewer #2: No

---

## [Author Response · Author response to Decision Letter 0]

16 Sep 2021

We thank the reviewers for their constructive comments. We have revised the manuscript to address all the questions and comments raised by the three reviewers. We highlight the changes made to the original version by setting the text color to red. Our specific responses to each comment are as follows:

Responses to reviewer #1: 

 “Quality of Life and cognitive assessment in healthy older Asian people with early chronic kidney disease: the NAHSIT 2013-2016 and validation study.”: Investigators intended to study the association between kidney function and cognition, as well as quality of life. The study is based on data from the NAHSIT dataset. Authors concluded that CKD stages 3 (a and b) is associated with cognitive decline in multiple domains, and that these patients have physical complaints.

The paper requires editing for English, for clarification of meaning.

• We are grateful for all the constructive comments. We have worked with an English editing service and have revised our manuscript accordingly.

Abstract

Purpose of the study is not stated.

• We revised the manuscript to address the purpose of the study: “We sought to obtain an overview and the attributable effect of impaired glomerular filtration on multiple domains in cognition and dimensions of quality of life for community-based healthy older adults in Taiwan. (Please see the Abstract section, page 2, Line 25-28)

Need to mention study is Taiwan-based.

• We revised the manuscript by mentioning that this study is a nationwide cross-sectional survey of older adults in Taiwan. (Please see the Abstract section, Page 2, Line 32).

In methods, need to mention this is data from NAHSIT 2013-2016.

• We added the description “The study data were derived from the Nutrition and Health Survey in Taiwan (NAHSIT) 2013–2016.” (Please see the Abstract section, Page 2, Line 29-30)

Also it says “…categorized into three CKD groups”… but they spell out 4 groups.

• We revised the text to say “Participants were categorized into four chronic kidney disease (CKD) groups.” (Please see the Abstract section, Page 2, Line 31).

Authors refer to those in stages 3 as “early CKD”; those would be patients with “moderate” disease.

• According to the definition by the National Kidney Foundation (NKF) Kidney Disease Outcomes Quality Initiative (KDOQI)[1] and the International Kidney Disease Improving Global Outcomes (KDIGO) guidelines [2], stages 3a (eGFR of 45–59 mL/min/1.73 m2), and 3b (eGFR of 30–44 mL/min/1.73 m2) of CKD indicate moderately impaired eGFR. 

• Stage 3 CKD (including 3a and 3b) are the first stages that could be identified by blood tests of creatinine alone. From the viewpoint of clinical management and policy level, a systematic review[3] concluded that Stage 3 CKD of mild to moderate impaired eGFR could be regarded as “early” CKD. A previous longitudinal study[4] showed that approximately half of patients with Stage 3 CKD progressed to late stages 4 and 5. Additionally, we found some relevant studies[5, 6] which defined Stages 3 as “early stages.” 

• In our revised manuscript, we followed the reviewer’s advice and designated Stage 3 as a “moderate” stage to improve readability. Accordingly, the title was revised as “Quality of Life and Cognitive Assessment in Healthy Older Asian People with Early and Moderate Chronic Kidney Disease: the NAHSIT 2013–2016 and Validation Study.” (Please see page 1, title; page 2, line 36, Abstract section)

In conclusion, “multiple cognitive decline”, “More attention for cognition should be paid for healthy older adults perceiving dissatisfaction on physical status”: unclear what these mean.

• In this study, we investigated both “quality of life” and “cognition” in older adults with early and moderate stages of CKD. We found that participants with stage 3 CKD manifested role-physical and vitality deficits in the dimensions of quality of life, and that participants with stage 3 CKD exhibited cognitive deficits in the domains of orientation to time, calculation, and complex commands. Therefore, participants had subjective deficits in role-physical behavior. To improve readability, the conclusion was revised as follows: “Elderly Han Chinese adults with moderately impaired renal filtration could manifest cognitive deficits in orientation to time, calculation, and impaired quality of life in physical-role functioning.” (Please see page 2, Line 43-45)

Introduction

-1st paragraph (prg): authors refer to USRDS annual report but don’t cite it.

• The citation of the reference “USRDS annual report 2020” was added[7]. (Please see page 4, line 53, Introduction section)

-The introduction is inadequate; the argument for this study is weak.

• In our revised manuscript, we substantiate the introduction by adding five relevant subsections as follows: (1) high prevalence of chronic kidney disease in Taiwan, (2) measures of chronic kidney disease, (3) cognitive impairment in chronic kidney disease, (4) quality of life in chronic kidney disease, and (5) study rationale and aims. In the first two sections, we accentuated the high prevalence of CKD in Taiwan and how kidney filtration rates were estimated in older adults. Thereafter, we summarized the current literature investigating the associations among cognitive impairment, quality of life, and worsened renal function. The study rationale has been revised accordingly. (Please see Page 3-6, Line 49-119, introduction)

-The specific aim/hypothesis need to be rewritten (in line with design) for clarity.

• We revised the hypotheses and specific aims as follows: We hypothesized that the high prevalence of cognitive impairment in older adults with early CKD in Taiwan, and the perception of the physical and mental well-being of older adults was worsened in the early stages of CKD. We aimed to (1) obtain an overview of cognitive impairment and quality of life in each stage of CKD, and (2) to assess the attributable effect of poor kidney filtration function on multiple cognitive domains dimensions of quality of life among community-based healthy older adults in Taiwan in this nationwide cross-sectional study. (Please see page 6, Lines 122-140, Introduction)

-Literature on cognition and CKD is not substantiated, despite the availability of important existing literature (such as with various tests of cognition performed on the Chronic Renal Insufficiency Cohort).

• In our revised manuscript, we substantiated the important existing literature. First, we introduced the pathophysiology of impaired renal filtration and cognition. Second, we reported that a vast majority of existing studies focused on reporting dementia in the late stages of CKD. Third, we listed the studies of systematic review and meta-analysis, which indicated that impaired eGFR is independently associated with increased MCI. Finally, we reviewed the relevant literature on investigating cognition in the early stages of CKD. (Please see Page 4-5, Line 73-101, introduction)

-Authors do not make the case for studying quality of life (QOL).

• In our revised manuscript, a new section for existing literature of “quality of life (QoL) and CKD” was added. We have cited and summarized these relevant studies, including a systematic review of QoL in late CKD stages, and the few available studies which enrolled participants in early CKD stages. In our review, whether quality of life was disturbed in the early stages of CKD remains undefined. There is also a lack of relevant research on ethnic Han Chinese. (Please see page 5, Line 103-119, introduction)

-Unclear why only “physical disability” and “caregiver burden” are highlighted.

• We revised our description to improve the readability of the study rationale and aims. In the first version of the manuscript, we emphasize that participants with early CKD may suffer from deficits in the perception of physical well-being and accentuate the caregiver’s burden. In addition, there is a high prevalence of CKD and ESRD and an increasing aging population in Taiwan. In our revised manuscript, the study rationale and aims have been rewritten to improve readability. (Please see page 6, Line 122-140, introduction)

Methods

-“systemic sampling”: do authors mean “systematic”?

• Thank you for your comment. We have revised the word to “systematic.” (Please see page 7, line 149, introduction).

-Authors need to clarify if this is a secondary analysis of data (2013-2016 of NAHSIT); otherwise, are they reporting primary data collection on MMSE/SF-12 as an add-on (to NAHSIT) conducted in those years?? This needs to be clear. What was the main aim of constructing the primary dataset?

• The study design and data collection in NAHSIT 2013–2016 were cited[8]. First, we declared this as a secondary analysis of original data from NAHSIT 2013 to 2016. The content of the primary survey included (1) behavioral indicators, such as MMSE and SF-12; (2) health outcome indicators, such as cardiovascular disease, kidney disease; and (3) laboratory tests of clinical chemistry, such as serum creatinine, lipid profiles, and blood glucose. Therefore, the MMSE and SF-12 data were collected during the study period 2013-2016 and these were not add-on variables. (Please see page 7, Line 151-157, Methods section)

• In addition, the primary aim of conducting NAHSIT by the Health Promotion and Administration, Ministry of Health and Welfare in Taiwan is to determine the nutritional status of the Taiwanese population, which was used as a reference for nutrition and health policy making in a government entity. (Please see page 7, Line 146-149, Methods section)

-Authors report this to be a “cross-sectional” study; need to clarify if all participant data were collected in only one visit.

• In our revised manuscript, we added the statement “All participants were visited once during the study period.” (Please see page 8, Line 155-156, Methods section)

-Data collection: clarify that this was for the primary study. Unclear how lab results on glucose and lipids, methodology on BP measurement are relevant in the context of this work. It would be helpful instead to elaborate on sociodemographic data for the current study.

• We clarified that data collection was for the primary objective of the study. (Please see page 8, Line 158, Method section)

• In our literature review, we found that diabetes increased the risk of cognitive impairment[9, 10] and an association between hyperlipidemia and mild cognitive impairment[11, 12].Therefore, we collected data on glucose and lipids, which were used for covariate adjustment in the multivariable regression models. 

• The only sociodemographic information available to us was educational level and individual income in the four CKD groups. In our analysis, we found no statistically significant differences in individual income among the four stages of CKD; however, years of education were significantly lower in the moderate and late stages of CKD. (Please see page 12, Lines 279-280, Results section; Table 1) 

-There is very limited description of measurement tools (MMSE, SF-12) pertinent to main study variables; additional details are needed such as SF-12 scoring and interpretation, applicability in Asian population, other.

• The culturally adapted traditional Chinese version of the Mini-Mental State examination (MMSE)[13] and SF-12 are the most widely used tools for assessing cognition and generic quality of life, respectively. We have added more detailed information regarding the measurement tools.

• The traditional Chinese version of the MMSE includes questions on orientation to time (5 points), orientation to place (5 points), registration (3 points), calculation (5 points), memory recall (3 points), language (2 points), repetition (1 point), and complex commands (6 points), and the MMSE score ranges from 0 (worst) to 30 (best).[14, 15] For the Taiwanese population, the norms[16] indicated 2 cutoff scores for determining cognitive impairment: a value of < 27 for literate individuals and another value of < 16 for the illiterate, respectively

• The most well-validated tool for quality of life assessment for the Taiwanese population is the traditional Chinese version of the SF-36[17, 18]. Despite being valid and equivalent for Han Chinese and other ethnic groups, SF-36 is limited by its length and time-consuming characteristics. The traditional Chinese version of SF-12, an abbreviated form of SF-36, is validated as equivalent to SF-36 for assessing the dimensions of physical and mental health.[19] The questionnaire of SF-12 questionnaire includes five dimensions of concepts: role-physical (RP), vitality (VT), social functioning (SF), role-emotional (RE), and mental health (MH). Each concept is directly transformed into a scale ranging from 0 to 100, with a mean distribution of 50 and a standard deviation of 10. A higher score signifies a better health state.[20, 21] (please see Page 7-8, Lines 165-185, Methods section)

-The use of Berlin Initiative Study equation for GFR estimation is not mainstream; authors did not elaborate on its applicability in Asian populations. Authors haven’t explained why those in stages 4 and 5 are grouped into one category. In labeling patients in stages 1 and 2 of CKD, how the diagnosis was made in addition to eGFR needs to be clarified.

• In our revised manuscript, we cited two studies[22, 23] which concluded that eGFR derived from the Berlin Initiative Study (BIS) equation achieved superior accuracy and lower rates of misclassification of CKD stages compared to the CKD-EPI and MDRD equations in older Chinese adults. These studies[22, 23] compared the different equations, including the Modification of Diet in Renal Disease (MDRD) equation, the Chronic Kidney Disease Epidemiology Collaboration (CKD-EPI) equation, and the BIS equation, to derive accurate eGFR (with 99mTc-DTPA as the standard method) in the Han Chinese population. (Please see page 9, Line 204-210, Methods section)

• In our revised manuscript, we added an explanation as to why those in stages 4 and 4 are grouped, as follows: “Since the NAHSIT 2013–2016 was expected to enroll healthy older adults, the number of late CKD Stages 4 to 5 was expected to be small, and therefore Stages 4 and 5 were grouped into one category.” (Please see page 9, Line 216-218, Methods section) 

• In addition to eGFR, we confirmed the early and moderate stages of CKD using the urine albumin-to-creatinine ratio (ACR). Impaired renal function allows albumin to pass into the urine. According to the international guideline group Kidney Disease Improving Global Outcomes (KDIGO) guidelines[2], urine ACR< 30 mg/g is defined as normal to mildly increased albuminuria, urine ACR of 30–300 mg/g as moderately increased albuminuria, and urine ACR >300 as severely increased albuminuria. (Please see Page 9-10, Line 219-222)

• The corresponding urine ACR values for all four groups are listed in Table 1. (Please see page 13-14, Table 1).

Statistical analysis: Elaboration is needed on the use of GLMM in this cross-sectional study. Authors need to clarify “covariates” used in the modeling, instead of referring to them as independent variables. The validation approach using principal component regression (factors 1 and 2, with dependent variable of eGFR) is unclear.

• We described this model in the following matrix formula Yi = Xi β + Ri + Ei, where Yi denotes the matrix containing a set of measures of all items of the MMSE and SF-12 on the ith older adult participant as dependent variables, Xi denotes the matrix containing a set of intercepts, independent variables of eGFR (per 30 mL/min/1.73 m2), and the covariates of age, sex, education years, and the characteristics that were imbalanced among the four CKD groups; β denotes the vector of estimated regression parameters; Ri denotes an intercept random effect associated with the ith older adult participant, which has a normal distribution and is independent and identically distributed (i.i.d.), where Ei denotes random error. (Please see page 10, Line 233-240, methods section)

• The covariates used in the GLMMs were none (model 1), a set of variables with age, sex, and interaction of age and sex (model 2), a set of variables with age, sex, and interaction of age and sex, and education years (model 3), and a set of variables with age, sex, and interaction of age and sex, education years, and the characteristics that were imbalanced among the four CKD groups (Model 4). (Please see page 10, Line 240-244, Methods section)

• Validation using principle component regression is elaborated as follows. To verify the association between renal function injury, cognitive impairment, and quality of life, we swapped the factors of eGFR in older adults and measured all items of the MMSE and SF-12 in the regression analysis. Conventionally, all items of the MMSE and SF-12 have been employed as dependent variables (or response variables), and eGFR has been used as an independent variable (or explanatory variable). Instead, we conducted partial least squares (PLS) regressions with the following formula: Y = XB + B0, where Y denotes the response matrix of eGFR, X denotes the factor matrix of all items of the MMSE and SF-12, B denotes the matrix of factor loadings, and B0 denotes error terms. Spearman’s rank correlation tests were used to examine the correlations among all items of the MMSE and SF-12. One major advantage of PLS regression is that it can easily manage the multicollinearity problem in conditions when there are numerous highly correlated explanatory variables[60, 61]. PLS regressions can help identify the optimal linear combinations of explanatory variables by projecting them to new spaces. The PLS regression was performed using the PROC PLS procedure implemented in SAS 9.4 software, with a factor number of 2 and principal components regression (PCR) to obtain factor loadings. (Please see page 11, Line 252-266, Methods section)

Results

-How was the 30-point change in eGFR coined in the study?

• According to the definition by the National Kidney Foundation (NKF) Kidney Disease Outcomes Quality Initiative (KDOQI)[1] and the International Kidney Disease Improving Global Outcomes (KDIGO) guidelines [2], CKD stage 1 was defined as eGFR >90 mL/min/1.73 m2, CKD stage 2 with eGFR of 60–89 mL/min/1.73 m2, and CKD stage 3 with eGFR of 30–59 mL/min/1.73 m2.

• Accordingly, the difference in eGFR between the early and moderate CKD stages is 30-point. A lower eGFR per 30 mL/min/1.73 m2 indicates a much worse renal function in the early and moderate stages of chronic kidney disease. In our revised manuscript, we have added this description to the footnotes of Table 3-4. (Please see Page 18-19, Table 3-4, Results section)

-In table 4, why does model 4 include HbA1C and HDL?

• In Table 1, we provide an overview of the characteristics of the four CKD stages. In our revised manuscript, we mentioned that in model 4, the characteristics or covariates that were imbalanced (P < 0.05), and the serum levels of HbA1c (P = 0.0430) and HDL-C (P = 0.0003) were not homogenous among the four groups of CKD; however, the mean HbA1c and HDL-C levels were within normal limits. (Please see Page 10, Line 244; Page 10, Methods; Page 13-14, Table 1, Results section)

-Validation study and results depicted in figures (depicting outcomes by age, sex)… are not in line with study purpose (there are no Specific Aims to support the conduct of those analyses).

• Our validation rationale is as follows: A validation study was used to examine the robustness of the association between eGFR, cognitive impairment, and quality of life. There was likely a high degree of collinearity in our data analysis, since multiple domains of cognition and multiple dimensions of life quality were assessed. We therefore replaced the conventional regression models with partial least squares (PLS) regressions to deal with potential collinearity and confirm robustness. (Please see Page 6, Line 135-140, Introduction; Page 11, Line 259-261, Methods section)

• Our aim was to use PLS regression to deal with possible collinearity. In our data analysis, we noticed weak collinearity of all items of the MMSE and SF-12. While all items in the MMSE were very weakly correlated, except for the weak association between calculation and complex commands (ρ = 0.374), all dimensions in SF-12 were weakly correlated, except for the moderate association between vitality and mental health (ρ = 0.566). (Please see Page 20, Line 341-344, Results section)

Discussion and Conclusion:

-The word “begin” would relate to longitudinal studies; its use should be reconsidered in this paper.

• In our revised manuscript, we deleted the words “begin,” and “launch” in the Discussion section. The description was revised as “early intervention for cognitive assessment should be conducted in CKD stages 3a and 3b in Han Chinese Older adults.” (Please see Page 21-22, Lines 363-373)

-Discussion/conclusion will require revision based on revisions in previous sections.

• Based on the literature review and analysis, our discussion and conclusions have been greatly revised. We addressed the following key characteristics: First, we enrolled a monoethnic Chinese population and we could manage this problem by employing the BIS equation. Our conclusion is consistent with previous studies[24-26] using the MDRD and CKD-EPI equations in different ethnic and age groups. Second, we adjusted the important covariates of demographic and sociodemographic factors and comorbidity in terms of laboratory tests. Third, we performed the analysis in both genders, unlike some studies with only men[27] and women[28, 29]. An earlier study in Taiwan[29] included only midlife women in Kinmen, a group of islands off the southeastern coast of mainland China, governed as a county in Taiwan. 

• Additionally, we addressed the limitation of employing the MMSE to assess cognitive function and premorbid conditions. The relevant discussion for quality of life in patients with CKD has been added. Lastly, we substantiated the discussion by adding possible behavioral interventions for CKD patients in further research. (Please see Page 22-25, Line 392-450)

-Unclear what is meant by “feeling physical disturbance” in conclusion.

• We are grateful for all the constructive comments. The conclusion was revised for clarity, as follows “More attention should be paid to healthy older adults with early stage of CKD having physical role function deficits in quality of life.” (Please see Page 25, Line 453-461)

 

Responses to reviewer #2:

The authors report on a large nationwide study of cognitive impairment and quality of life in CKD patients 65 years and above. There are a couple of areas that can be improved, particularly in terms of study rationale, terminology and phrasing of interpretations, which currently reduce my enthusiasm for this manuscript.

• We are grateful for all the constructive comments. In our revised manuscript, we have greatly revised the study rationale, terminology, and phrasing. 

• In terms of rationale, Taiwan has the highest worldwide prevalence of CKD and ESRD [7]. Therefore, assessment of cognition in CKD is of paramount importance, and we hypothesized that the high prevalence of cognitive impairment in older adults with CKD in Taiwan, and perception of the physical and mental well-being of older adults was worsened in the early stages of CKD. We aimed (1) to obtain an overview of cognitive impairment and quality of life in each stage of CKD, and (2) to assess the attributable effect of poor kidney filtration function on multiple domains or dimensions of cognition and quality of life among community-based healthy older adults in Taiwan in this nationwide cross-sectional study. (Please see page 6, Line 122-140)

Line 31-32: Merge CKD stages 1 and 2 to match the statement that 3 groups were derived. At present it reads as 4 separate groups.

• Thank you for your comment. We revised the description into four groups: (1) CKD stage 1, (2) CKD stage 2, (3) CKD stages 3a and 3b, and (4) CKD stages 4 and 5. We merged CKD stages 4 and 5 into a single group due to the smaller sample size of late CKD stages in our community study. (Please see page 2, line 31).

The Introduction itself is short and lacking on details that contribute to a well-supported study rationale. In particular, the authors have not provided any evidence from previously published papers showing that cognitive impairment is related to eGFR levels (see Etgen T et al Chronic kidney disease and cognitive impairment: a systematic review and meta-analysis. Am J Nephrol. 2012;35(5):474–82), which would support the notion that cognitive impairment would be present in early as well as late stage CKD. In fact, this is evidenced in the Berger et al. 2016 systematic review and meta-analysis which the authors cite in the Discussion. Furthermore, in another systematic review (Brodski, J. et al. (2019). A Systematic Review of Cognitive Impairments Associated With Kidney Failure in Adults Before Natural Age-Related Changes. Journal of the International Neuropsychological Society, 25(1), 101-114.), it has been shown that the cognitive impairments found in relation to CKD are independent of age-related cognitive decline, and incrementally increase as the stages of CKD advance. At the very least, these key reviews/meta-analyses should be cited to support the argument for the authors investigating cognition in early-stage CKD, even if within and despite an older cohort.

• In our revised manuscript, we have expanded the contents of the Introduction section. We substantiated the content by adding five relevant subsections as follows: (1) high prevalence of chronic kidney disease in Taiwan, (2) measures of chronic kidney disease, (3) cognitive impairment in chronic kidney disease, (4) quality of life in chronic kidney disease, and (5) study rationale and aims. 

• Accordingly, these important studies have been cited in our manuscript. We cited two meta-analyses by Etgen et al.[24] and Berger et al.[25] showed that impaired eGFR is associated with cognitive impairment and can be present in the early and late stages of CKD. A systematic review[26] included three relevant studies on CKD patients <65-year-old which found that these patients had slow processing speed[30], impaired verbal learning[29, 31], and impaired working memory[29] in moderate stages of CKD. Our argument has been revised. (Please see Page 3-6, Lines 48-140, introduction)

The Introduction is also lacking in any background information to support an investigation of quality of life within a CKD sample. Please add this information.

• We amended the introduction section and added background information to support an investigation of the quality of life of older Chinese adults with CKD as follows:

• The term “quality of life” can be traced back to the definition in 1948 by the World Health Organization (WHO) as a state of perception of “physical, mental, and social well-being, and not merely the absence of disease.[32]” Prior meta-analysis[33] assessing the utility-based quality of life with 12-item or 36-item Short Form Health Survey (SF-12 or SF-36), EuroQol Group’s EQ-5D, and other instruments suggested that dialysis is significantly associated with reduced quality for ESRD patients. However, few studies have investigated the association between quality of life and the early stages of CKD. We found that these studies exhibited inconsistent results: (1) a cross-sectional study enrolled ≥60 year-old participants in North California[34] showed no relationship between eGFR and quality of life scores in physical and mental dimensions after adjustment for medical comorbidities; (2) a cross-sectional study including ≥50 year-old Irish participants [35] showed that reduced scores in quality of life are associated with eGFR by serum cystatin rather than creatinine; and (3) a recent cross-sectional study[36] enrolled 2,255 older adults in European countries and found that early CKD at stages 3a and 3b, defined by the BIS equation, is associated with quality of life on EQ-Visual Analogue Scale (EQ-VAS). To our knowledge, there is a shortage of relevant research in the Chinese population. (Please see Page 5, Line 103-120, introduction section)

Based on the existing information in the literature, once added to the Introduction, the authors should revise the hypotheses to be a bit more specific. I would also recommended a statement of aims to cover the components of the proposed analyses.

• We revised the introduction section by adding existing literature investigating the association between cognition and CKD, and between quality of life and CKD. The specific aims of this study are as follows: 

• In this study, we hypothesized that the high prevalence of cognitive impairment in older adults with CKD in Taiwan, and perception of the physical and mental well-being of older adults was worsened in the early and moderate stages of CKD. We aimed to (1) obtain an overview of cognitive impairment and quality of life at each stage of CKD in Han Chinese population, and (2) assess the attributable effect of poor kidney filtration function, after adjustment for demographic factors, on multiple cognitive domains and dimensions of quality of life among community-based healthy older adults in Taiwan in this nationwide cross-sectional study. (Please see page 6, Line 122-140, introduction section)

The authors are further requested to clarify and provide a rationale for the validation study within the Introduction section. At present, it appears abruptly.

• The rationale for conducting the validation study was as follows: The existence of collinearity in our data analysis was likely, since multiple domains of cognition and multiple dimensions of life quality were assessed. As a separate validation study, we employ partial least squares (PLS) regressions to solve potential collinearity and confirm the robustness of our results. (Please see Page 6, Line 135-140)

Was information about co-morbid conditions (e.g. diabetes, cardiovascular disease, neurological disorders/conditions)? There is a sizeable literature showing cognitive impairments associated with these conditions, which need to be considered for the current dataset. If this information was not collected, it is a major limitation of the study and needs to be mentioned in text.

• In our study, we assessed comorbid diabetes by measuring fasting glucose and HbA1c levels. According to the American Diabetic Association[37], participants were classified as having DM by fasting glucose ≥126 mg/dl or HbA1c level ≥6.5%. 

• Hypertension is the only cardiovascular disease variable available to us. We defined participants with hypertension as systolic blood pressure systolic BP (SBP) of ≥140 mmHg or diastolic BP (DBP) of ≥90 mmHg, in accordance with the guidelines of the American Heart Association[38]. Otherwise, cardiovascular and neurological diseases were not assessed in the NAHSIT 2013–2016 (Please see Page 9, Line 197-201). We also mentioned this limitation in our discussion section. (Please see Page 24, Line 435-438)

Given the number of analyses conducted, correction for multiple comparisons (e.g., Bonferroni correction) in the group comparisons is required to minimise Type 1 error.

• Applying the Bonferroni correction formula p value ≤ α/m, where the significance level is α, and m is the number of hypotheses for multiple comparisons. In our study, we defined α as 0.05 and four groups (or hypothesis) comparisons were conducted. Applying the Bonferroni correction, a two-tailed p value of < 0.0125 (0.05/4) was considered statistically significant. (Please see Page 10-11, Line 245-248, method)

Participant group characteristics could be reported in a table for greater clarity and ease of comprehension.

• In our revised manuscript, we sorted the group characteristics into six categories for clarity. The six categories were as follows: demographic characteristics (age and sex), sociodemographic status (education years and individual income), kidney function measures (including blood urea nitrogen, creatinine, eGFR, urine albumin-to-creatinine ratio), comorbid conditions (including hypertension, hyperlipidemia, diabetes), blood pressure, and serum laboratory studies (including fasting glucose, HbA1c, total cholesterol, LDL-C, HDL-C, and triglycerides). (Please see Page 13-14, in Table 1).

It is not appropriate to say that MMSE scores decreased between stages as the participants were not followed up longitudinally. As what has been conducted are just group comparisons, the differences should be described as performance differences between groups of patients at the different stages. The use of ‘decreasing’ is misleading in this context of reporting group comparison results. In addition, within the adjusted GLMMs, it should also be phrased similarly (e.g. lower eGFR is associated with reduced QOL). Please amend throughout.

• In our revised manuscript, we replaced the word “decrease” with “lower” to avoid misleading. Accordingly, the descriptions in the Abstract, Results, and Tables 3-4 were revised.

In line 205, please refrain from using terminology such as ‘thoroughly investigated’ when many cognitive domains were not assessed in this study.

• Thank you for your comment. We deleted the word “thoroughly” in the text. (Please see page 21, line 363).

In lines 210-211, it is not accurate to say that “patients of late CKD stages 4–5 with poorer orientation, executive function begin in the early stage of CKD (stages 3a and 3b)” as the authors did not longitudinally follow a group of patients as their CKD progressed. I believe the authors are suggesting that executive function deficits are present earlier in the illness at Stage 3, so it would be more accurate to just say that executive function deficits are evident from Stage 3 CKD.

• In our revised manuscript, we have amended the description as follows: The study results expand the previous findings[25, 39] that executive function deficits are evident earlier in the CKD illness at stage 3, and patients with late CKD stages 4–5 with poorer orientation in the older Han Chinese population. (Please see page 21-22, Line 367-373, Discussion section)

In line 220, increasing age and female sex cannot promote cognitive impairment, rather they can be associated with greater cognitive impairment. Please rephrase.

• We revised this description as “demographic contribution should not be a single factor to explain cognitive impairment in our participants with early stage CKD. After adjustment for age and sex in models 2–4, the GLMMs showed that lower eGFR was independently associated with cognitive impairment.” (Please see page 20, line 377-380, Discussion section)

In line 224, it would be better to say that clinical factors do not really explain the differences in cognitive impairment between the groups.

• Thank you for your comment. We amended this text as “clinical factors did not explain the differences in cognitive impairment among the CKD groups.” (Please see page 20, Line 380-381, Discussion section)

Line 230-232: It is not intuitive how the authors arrived at the conclusion that their analysis “analysis supported the association between early renal function impairment and cognitive impairment should not be the confounding effect of aging only” based on the statement preceding it. Please clarify. In addition, such a conclusion has previously been reached in the literature (Brodski et al, 2019). Relatedly, the authors should better integrate and situate their findings in relation to the existing literature.

• We revised the conclusion here to say, “our analysis supported the presence of cognitive impairment in early to moderate stages of CKD in Han Chinese older adults.”

• Compared to previous studies, our study has the following key characteristics: First, previous analysis did not clarify which is the best method for assessing eGFR in elderly and cognitively impaired adults.[24, 40] Since our enrolled population is a monoethnic Chinese population, we could manage this problem by employing the BIS equation. Our conclusion is consistent with previous studies[24-26] using the MDRD and CKD-EPI equations in different ethnic and age groups. Second, our analysis included covariates of demography, sociodemographic factors, and medical comorbidities in terms of laboratory tests. Third, we performed the analysis in both genders, unlike some studies with only men[27] and women[28, 29]. An earlier study in Taiwan[29] included only midlife women in Kinmen, a group of islands off the southeastern coast of mainland China, governed as a county in Taiwan. Fourth, our nationwide sampling should have a representative population of Han Chinese. We performed a comprehensive door-to-door survey conducted by trained interviewers and laboratory examinations. Fifth, we validated the study results using the unconventional method of partial least squares regressions with principal components to investigate the factor loadings on the response variable of eGFR. Factor loadings on orientation, calculation, and complex domains were all > 0.3, which is consistent with the analysis of GLMMs. Sixth, both cognition and quality of life were surveyed in our study. (Please see Page 23-24, Line 405-421, Discussion section)

Line 234-242: It is unclear how this paragraph fits within the narrative.

• We have removed this paragraph from the Discussion section. Instead, the reasons for using generalized linear mixed models (GLMMS) were added in the Methods section. 

Regarding limitations, the authors should also discuss the limits of using the MMSE for assessing cognitive function within CKD (see Brodski et al, 2019). Also see my earlier point about premorbid conditions.

• In our revised manuscript, we added the limitation of using the MMSE to assess cognitive function and premorbid conditions. These were addressed as follows:

• Despite being the most widely used instrument in the extant literature, the MMSE is known to be affected by ceiling and floor effects in different languages and populations.[26, 41, 42] While the ceiling effect indicates that the performance of MMSE, independent of cognitive state, is favorably affected by a high level of education, the floor effect suggests that performance is adversely affected by poor education.[41] In addition, the MMSE is known to be less sensitive to mild changes in cognition, and therefore may have reduced sensitivity for early CKD stages.[26] These facts could limit the generalizability and comparability of similar studies. Some studies[43, 44] indicated that the Montreal Cognitive Assessment (MoCA) showed better utility as a tool for assessing patients with CKD. Third, we did not assess severe cardiovascular diseases or other neurological disorders in our study. 

• Since the NAHSIT 2013–2016 primarily focused on healthy older adults in adults, only health comorbidities of hypertension, hyperlipidemia, and diabetes mellitus were assessed in our study. (Please see Page 24, Line 423-438, Discussion section)

The authors should also briefly discuss the utility of cognitive remediation to address cognitive deficits, particularly in early CKD (see Tan, E. J. et al (2019). Considering the utility of cognitive remediation therapy in chronic kidney disease. Clinical and Experimental Nephrology.) for some points. This would help better contextualise the current findings.

• Thank you for your comment. The relevant discussion of cognitive remediation for older adults with early stages has been added to the manuscript as follows: Our study confirmed that CKD with cognitive impairment (especially the domains of orientation, calculation, and complex commands) is present in the early to moderate illness stages. A non-invasive behavior intervention by cognitive remediation (CR) is proposed to help promote neuroplastic change and enhance cognitive performance in patients with CKD [45]. Different CR approaches[46] can be tailored to specific cognitive deficits in patients with CKD. However, there is a shortage of CR intervention research studies to date.[45] (Please see page 24-25, Lines 440-450, Discussion section)

Please review the manuscript for grammatical errors throughout.

• Thank you for your comment. We consulted an English editing service to improve readability and grammatical errors in our revised manuscript.

---

## [Decision Letter · Decision Letter 1]

21 Feb 2022

Quality of Life and Cognitive Assessment in Healthy Older Asian People with Early and Moderate Chronic Kidney Disease: the NAHSIT 2013–2016 and Validation Study

PONE-D-20-31145R1

Dear Dr. Bai,

We’re pleased to inform you that your manuscript has been judged scientifically suitable for publication and will be formally accepted for publication once it meets all outstanding technical requirements.

Kind regards,

Abduzhappar Gaipov

Academic Editor

PLOS ONE

Additional Editor Comments (optional):

Authors had responded to all required comments

Reviewers' comments:

Reviewer's Responses to Questions

**Comments to the Author**

1. If the authors have adequately addressed your comments raised in a previous round of review and you feel that this manuscript is now acceptable for publication, you may indicate that here to bypass the “Comments to the Author” section, enter your conflict of interest statement in the “Confidential to Editor” section, and submit your "Accept" recommendation.

Reviewer #3: All comments have been addressed

2. Is the manuscript technically sound, and do the data support the conclusions?

Reviewer #3: Yes

3. Has the statistical analysis been performed appropriately and rigorously? 

Reviewer #3: Yes

4. Have the authors made all data underlying the findings in their manuscript fully available?

Reviewer #3: Yes

5. Is the manuscript presented in an intelligible fashion and written in standard English?

Reviewer #3: Yes

6. Review Comments to the Author

Reviewer #3: (No Response)

7. PLOS authors have the option to publish the peer review history of their article (what does this mean?). If published, this will include your full peer review and any attached files.

Reviewer #3: No

---

## [Editor Report · Acceptance letter]

28 Feb 2022

PONE-D-20-31145R1 

Quality of Life and Cognitive Assessment in Healthy Older Asian People with Early and Moderate Chronic Kidney Disease: the NAHSIT 2013–2016 and Validation Study 

Dear Dr. Bai:

I'm pleased to inform you that your manuscript has been deemed suitable for publication in PLOS ONE. Congratulations! Your manuscript is now with our production department. 

Kind regards, 

on behalf of

Dr. Abduzhappar Gaipov 

Academic Editor

PLOS ONE